# PLA: The Optimal Path from Softmax Attention to Linear Models via KV Cache Compression

## Abstract

Transformers, despite their remarkable sequence modeling capabilities, are fundamentally constrained by the quadratic complexity of Softmax attention and the unbounded growth of the key–value (KV) cache. Replacing Softmax attention with linear variants has emerged as a promising direction, yet existing approaches lack a systematic functional comparison with Softmax attention, clear error analysis, and a theoretically guided roadmap for improvement. In this work, we approach the problem from the perspective of KV cache compression and present a theoretically grounded pathway from Softmax attention to linear models. Our analysis reveals five critical components: redundancy elimination, tokenizer-level quantization and positional information separation, positional information compression, inter-layer similarity, and multi-state decomposition. For each, we provide succinct theoretical justification, derive error bounds, and demonstrate equivalence to existing mechanisms. Building on this pathway, we introduce PLA, a linearized attention model that inherits pretrained weights and achieves state-of-the-art performance. Notably, PLA surpasses strong baselines such as MVA and GSA on multiple benchmarks while requiring only 80% of the fine-tuning resources. Our findings provide both theoretical clarity and practical guidance for advancing linear attention, highlighting a principled route towards efficient and scalable alternatives to Softmax attention.

## 1 Introduction

The Transformer architecture Vaswani et al. (2017) has become the backbone of modern deep learning, powering state-of-the-art models in language Touvron et al. (2023); Jiang et al. (2023); DeepSeek-AI et al. (2024); Dubey et al. (2024); Yang et al. (2025), vision Dosovitskiy et al. (2020); Han et al. (2022), and multimodal Yin et al. (2024) domains due to its remarkable sequence modeling capability. Despite these successes, Transformers face two fundamental limitations: the *quadratic complexity* of Softmax attention with respect to sequence length, and the *unbounded growth* of the key-value (KV) cache during autoregressive inference. These issues severely constrain the applicability of Transformers in long-sequence modeling tasks, such as video understanding Tang et al. (2025), genomic sequence analysis Jumper et al. (2021), and other domains requiring extended context Jiang et al. (2025).

To address these limitations, two major lines of research have emerged: *KV cache compression* Luohe et al. (2024); WEI et al. (2025) and *linear attention* Katharopoulos et al. (2020); Hua et al. (2022); Qin et al. (2024a). KV cache compression methods aim to reduce memory usage by compressing the stored states across either the sequence or the channel dimension. For example, some approaches design task-specific prompts that select a fixed set of relevant KV entries to retain, thus improving efficiency at the cost of generality. Others Hu et al. (2021) apply low-rank or dimensionality reduction techniques along the channel dimension, yielding constant compression ratios but failing to fundamentally address the ever-growing cache size in long contexts.

Linear attention methods, in contrast, replace the Softmax kernel with kernelized approximations, thereby reordering computations to achieve linear complexity. Crucially, such models can recursively maintain a fixed-size state, resolving the KV cache growth problem. Recent works Yang et al.

(2024b;a) further enhance these models with additional mechanisms such as gating functions and delta-rule updates to improve expressiveness. However, linear attention models still exhibit significant drawbacks: they often suffer from limited retrieval and reasoning capacity, exhibit noticeable performance gaps relative to Softmax attention, and typically require training from scratch or hybridization with Softmax attention to achieve competitive results.

In this work, we revisit the connection between Softmax attention and linear attention from the perspective of *KV cache compression*. We propose what we argue to be the current optimal and theoretically grounded pathway for compressing Softmax attention into linear models. This pathway is structured around five theoretical principles, each demonstrating (i) the necessity of a specific compression step, (ii) its equivalence to mechanisms in existing approaches, and (iii) the error it introduces relative to Softmax attention. Taken together, these principles provide a clear functional blueprint of what linear attention should retain, what it can safely discard, and how it differs fundamentally from Softmax attention.

Our analysis yields both theoretical and practical benefits. First, it clarifies the essential components required to bridge the gap between Softmax and linear attention, guiding future designs of efficient architectures. Second, it enables the transformation of pretrained Softmax-based large language models into linear variants with significantly reduced fine-tuning cost. Empirically, we show that our approach achieves state-of-the-art performance, narrowing the gap between linear and Softmax attention especially on tasks where existing linear models struggle, such as retrieval, few-shot reasoning, and complex logical inference.

## 2 BACKGROUND AND PRELIMINARIES

### 2.1 TRANSFORMERS AND SOFTMAX ATTENTION

The Transformer architecture relies on the attention mechanism to dynamically compute contextualized representations. Given an input sequence $\boldsymbol{X} \in \mathbb{R}^{t \times d}$, it is linearly projected into queries $\boldsymbol{Q}$, keys $\boldsymbol{K}$, and values $\boldsymbol{V}$. Attention is then computed as

$$\boldsymbol{O} = \text{Attention}(\boldsymbol{Q}, \boldsymbol{K}, \boldsymbol{V}) = \text{Softmax}\left(\frac{\boldsymbol{Q}\boldsymbol{K}^\top}{\sqrt{d}} \odot \boldsymbol{M}\right)\boldsymbol{V}, \boldsymbol{o}_i = \sum_{j=1}^{i} \frac{\exp\left(\frac{\boldsymbol{q}_i \boldsymbol{k}_j^\top}{\sqrt{d}}\right)}{\sum_{h=1}^{i} \exp\left(\frac{\boldsymbol{q}_i \boldsymbol{k}_h^\top}{\sqrt{d}}\right)} \boldsymbol{v}_j, \quad (1)$$

where $\boldsymbol{M}$ denotes the causal mask with $\boldsymbol{M}_{ij} = 1$ if $i \geq j$ o.w. $-\infty$ and $d$ is the feature dimension used for normalization. Equivalently, the autoregressive form can be written as $\boldsymbol{o}_i$, where $\boldsymbol{q}_i$, $\boldsymbol{k}_j$, and $\boldsymbol{v}_j$ are the $i$-th or $j$-th row vectors of $\boldsymbol{Q}$, $\boldsymbol{K}$, and $\boldsymbol{V}$, respectively. While highly effective, this formulation entails quadratic complexity in sequence length and requires storing all past key–value pairs, leading to unbounded KV cache growth during inference.

### 2.2 KV CACHE COMPRESSION METHODS

To alleviate the quadratic growth of the key–value (KV) cache, a large body of work explores *KV cache compression*. The central idea is to reduce redundancy in the cache by performing low-rank transformations or selection operations along the sequence dimension or the channel dimension.

Formally, given an input $\boldsymbol{X} \in \mathbb{R}^{t \times d}$ and its projections $\boldsymbol{Q}, \boldsymbol{K}, \boldsymbol{V} \in \mathbb{R}^{t \times d}$, the compressed cache $(\boldsymbol{K}^c, \boldsymbol{V}^c)$ of size $c \times d^r$ is defined as

$$\boldsymbol{K}^c = \varphi\big(\phi(\boldsymbol{R}\boldsymbol{K})\boldsymbol{L}\big), \qquad \boldsymbol{V}^c = \varphi\big(\phi(\boldsymbol{R}\boldsymbol{V})\boldsymbol{L}\big), \tag{2}$$

where $\boldsymbol{R} \in \mathbb{R}^{c \times t}$ selects $c$ tokens from the sequence, and $\boldsymbol{L} \in \mathbb{R}^{d \times d^r}$ compresses the channel dimension. Here $d^r = h \times d_k^r$, with $h$ denoting the number of heads and $d_k^r$ the per-head compressed dimension. The operators $\phi(\cdot)$ and $\varphi(\cdot)$ denote transformation and selection functions, respectively.

Because the sequence length $t$ grows without bound during autoregressive generation, $\boldsymbol{R}$ is usually constructed recursively, i.e.,

$$\boldsymbol{R} = f(\boldsymbol{R}'\boldsymbol{X}^\top) \in \mathbb{R}^{c \times t}, \tag{3}$$

where $\boldsymbol{R}' \in \mathbb{R}^{c \times d}$ defines a local observation window, and $f(\cdot)$ specifies the selection strategy.

(1) **SnapKV** Li et al. (2025) applies compression only along the sequence dimension. Specifically, $L$ is the identity matrix, so no channel compression is applied. $R = f(QK^\top)$ is defined via a $\texttt{top-}c$ operator over the most recent queries and obtains the indexes of the corresponding KV block. $\phi(\cdot)$ gather the corresponding key tokens by these indices. The same procedure is applied to $V$. (2) **HeadKV** Fu et al. (2025) extends SnapKV by compressing along the channel dimension. In this case, $R$ again selects tokens as in SnapKV, while $L$ is defined through an additional projection $L'$, which serves as a voting mechanism across attention heads. $\varphi(\cdot)$ gather the corresponding key heads by these votes. (3) **Multi-Head Latent Attention (MLA)** DeepSeek-AI et al. (2025) approach removes explicit token selection. Both $\phi(\cdot)$ and $\varphi(\cdot)$ are set to the identity. No sequence compression is applied, i.e., $R = I$. Instead, channel compression is performed with a fixed projection $L = (W^{UK})^{-1}$. By leveraging the associativity of matrix multiplication to fuse the up-projection with the query-key product, MLA avoids explicitly reconstructing $K$ and $Q$, achieving substantial memory savings and computational speedup.

Numerous subsequent approaches, including InfLLM Xiao et al. (2024a), HO2 Zhang et al. (2023), and StreamLLM Xiao et al. (2024b), can be understood as hybrids or equivalent reformulations of the above principles.

## 2.3 LINEAR MODELS

Linear attention replaces the Softmax kernel with linearizable feature maps, which permits re-ordering the computations among queries, keys and values and thereby achieves linear time and fixed-size state:

**Parallel form.**
$$O = \mathrm{LA}\big(\phi(Q), \phi(K), V\big) = \big((\phi(Q)\phi(K)^\top) \odot M\big) V, \qquad (4)$$

**Recursive form.**
$$S_t = S_{t-1} + \phi(k_t)^\top v_t, \qquad o_t = \phi(q_t)\, S_t, \qquad (5)$$

where $S_t \in \mathbb{R}^{d_k \times d_v}$ is a fixed-size state matrix maintained across time steps. By keeping $S_t$ bounded, linear attention attains constant memory during autoregressive inference. Many works focus on improving the choice of $\phi(\cdot)$ Han et al. (2023); Choromanski et al. (2022) or introducing auxiliary mechanisms to enhance expressiveness.

However, this formulation is prone to state saturation, which dilutes the attention mechanism. To address this, methods like GLA introduce a gating mechanism that enables dynamic forgetting in the state $S_t$, thereby promoting a bias towards more recent context.

**Gating / GLA.** Gated Linear Attention (GLA) Yang et al. (2024b) applies multiplicative gates to control the contribution of new tokens and the persistence of prior state:

$$O = \mathrm{GLA}(Q, K, V, G) = \mathrm{LA}(Q \odot B,\ K/B,\ V), S_t = \mathrm{diag}(g_t)\, S_{t-1} + k_t^\top v_t, o_t = q_t\, S_t. \qquad (6)$$

where the $t$-th row of $B$ is $b_t = \prod_{i=1}^{t} g_i$ and $G = \sigma(XW_g) \in \mathbb{R}^{n \times d_k}$. While models like MetaLA Chou et al. (2024), HGRN2 Qin et al. (2024b), and GSA Zhang et al. (2024) also employ gating, this approach offers a relatively coarse control over the state $S_t$, failing to fully address information redundancy. This limitation motivated the development of more sophisticated updates, such as those inspired by fast weights.

**Fast-weight / Delta Rule-style updates.** These methods aim to correct the stored representation based on prediction error:

$$v_t^{\mathrm{old}} = k_t\, S_{t-1}, \qquad S_t = S_{t-1} + g_t \cdot k_t^\top \big(v_t - v_t^{\mathrm{old}}\big), \qquad o_t = q_t\, S_t. \qquad (7)$$

Intuitively, the update adds a correction proportional to the discrepancy between the newly observed value $v_t$ and the state-predicted value $v_t^{\mathrm{old}}$, gated by $g_t$. Building on this, DeltaNet Yang et al. (2024a) parallelized the formulation to create a powerful linear model.

**Optimization viewpoint.** Furthermore, viewing the update through an optimization lens led to dynamic weighting methods like TTT Sun et al. (2025), Titans Behrouz et al. (2024), and Atlas Agrawal et al. (2025), which share the unified objective:

$$\mathcal{L}_t(M) = \sum_{i=1}^{t} \gamma_i \big\| M\big(\phi(k_i)\big) - v_i \big\|_2^2, \qquad (8)$$

where $M(\cdot)$ denotes a (possibly parametric) mapping from key-features to value-predictions and $\{\gamma_i\}$ are weighting coefficients. One may then update the memory/map $M_t$ by performing a few steps of iterative optimization (e.g., gradient descent-GD or Muon Jordan et al. (2024)-style). A compact schematic of such an optimization-inspired update is:

$$M_t = \alpha_t\, M_{t-1} + F(S_t), \quad S_t = \eta_t\, S_{t-1} - \theta_t\, \nabla_S\, \mathcal{L}_t\big(M_{t-1}; k_t, v_t\big), \tag{9}$$

where $F(\cdot)$ aggregates the current statistics into the primary memory $M_t$, and the second line denotes a gradient-based (or similar) corrective step for the working state $S_t$. This optimization viewpoint explains a number of empirically successful update rules and motivates algorithms that explicitly minimize per-step predictive error.

**Extending the state budget.** Even with sophisticated updates, a single low-rank state may remain insufficient to capture complex, multi-scale dependencies. Recent works therefore maintain multiple parallel or hierarchical states, each specialized for different temporal ranges or functional roles. For instance, **MoM** Du et al. (2025) and **MVA** Wang et al. (2025) maintain multiple memory banks (e.g., short-term vs. long-term) and/or decompose the state into several sub-states that interact during read/write.

$$S_t^{(i)} = \mathrm{diag}(1 - \bar{f}^{(i)}(x_t^{(i)})^\top)S_{t-1}^{(i)} + \bar{f}^{(i)}(x_t^{(i)})^\top x_t^{(i)}, \qquad x_t^{(i+1)} = f_1^{(i)}(x_t^{(i)}, S_t^{(i)}) \tag{10}$$

where $f$ function is generally taken as $\sigma$, while the $f_1$ function is taken as a hybrid expert or delta function. This multi-state design can close much of the performance gap to Softmax attention, at the cost of additional architectural and algorithmic complexity.

## 3 METHOD

Existing KV cache compression methods either lack general applicability or fail to address the unbounded growth of KV cache, while also lacking a clear error analysis compared to Softmax Attention. Furthermore, current linear attention approaches lack a comprehensive understanding of their components and mechanisms from the perspective of Softmax Attention, resulting in the absence of clear improvement strategies to match or even surpass Softmax Attention performance.

To address these limitations, this paper presents five theoretical principles with corresponding experimental validation, establishing an optimal pathway for compressing Softmax Attention into linear attention. Each theoretical node provides rigorous error analysis and demonstrates equivalence to existing model operations and mechanisms, thereby offering valuable references and guidance for future improvement strategies.

### 3.1 NECESSITY OF REDUNDANCY REMOVAL

**Theorem 1** (Necessity of Redundancy Removal). *For any sequence stored in the KV cache that exhibits translation-invariant properties, the application of redundancy removal operations (e.g., unique filtering) together with a counting mechanism can bound the storage size. Specifically, the upper bound is given by*

$$C \le 2^{b_k \times d_k + b_v \times d_v} \times \big(b_k \times d_k + b_v \times d_v + b_c\big), \tag{11}$$

*where only the unique KV vectors $u_i$ and their counts $c_i$ are stored as a set of unique pairs $\{(u_i, c_i)\}_{i=1}^{C}$, with $i \le 2^{b_k \times d_k + b_v \times d_v}$.*

*Here $b$ denotes the bit-width of the number type for the KV vectors (e.g., $b = 16$ for* `float16`, *$b = 4$ for* `int4`*), $d_k$ and $d_v$ are the dimensions of the key and value vectors (e.g., head_dim $= 128$ in Qwen and LLaMA), and $b_c$ is the number of bits required to maintain counts. This compression is lossless, and the redundancy-removal mechanism is functionally equivalent to the Delta Rule used in existing fast-weight models.*

A detailed proof is provided in the Appendix A.1.

**Generalization.** The unique operation can also be generalized by relaxing exact matching to cosine similarity. For example, using a threshold of $0.9$ instead of $1.0$ can still preserve performance

in A.1. This opens the possibility of balancing efficiency and accuracy by tuning the threshold. Furthermore, we propose a more general and stronger compression approach: when two tokens exhibit cosine similarity above a threshold, we treat them as identical and replace them by their average. This can be interpreted as a quantization process, which also provides noise reduction.

**Complexity Implication.** Let the upper bound of the KV cache be $C$. According to the above reasoning, Softmax attention can be interpreted as a linear model with complexity $\mathcal{O}(N \times C \times d)$, where $C = 2^{2bd}$. Since $C$ is extremely large, the naive bound is impractical in comparison to specialized task-optimized methods. Nonetheless, the key insight here is that redundancy removal (via unique or Delta Rule operations) is *necessary* to compress an unbounded state into a bounded one.

## 3.2 Tokenization and Positional Information Decoupling

To achieve stronger compression, we conduct a deeper analysis of sequence modeling in LLMs. The input to an LLM undergoes tokenization, which constitutes a strong quantization that limits the number of distinct types to the vocabulary size $V_T$ (e.g., 32K for LLaMA2). Subsequent channel mixing operations in the LLM do not affect the number of distinct types; rather, it is the positional encoding and token mixing operations that impact type diversity. We therefore optimize the input-output characteristics of these two operations, leveraging the first-layer tokenization to achieve stronger compression and reduce the upper bound of the state storage requirement.

We introduce two optimizations specifically targeting positional encoding operations:

**Theorem 2** (Necessity of Positional Information Decoupling). *For tokenizer-based models, the input sequence and KV cache at the first layer, after positional information decoupling, can be losslessly represented with an upper bound of $C = V_T$, where $V_T$ is the vocabulary size. This is achieved by storing a tensor of size $V_T$ along with indices for each vector. The limitation of this approach is that the indices grow unbounded with sequence length, theoretically only compressing storage by a factor of the dimension size.*

Due to positional constraints, the upper bound remains identical to Theorem 1. We subsequently address this limitation through positional information compression. The detailed proof is provided in Appendix A.2. This approach provides lossless compression for the first layer.

## 3.3 Positional Information Compression

Building upon the upper bound established in Theorem 2, we further optimize the positional information representation to achieve the same upper bound as the position-agnostic KV cache in Theorem 2. This allows us to completely control the first-layer upper bound at the vocabulary size level, laying the foundation for fully fixed-size state linear models.

**Theorem 3** (Necessity of Positional Information Compression). *With the compression method of positional encoding described below, the upper bound of the KV cache and the positional information in the first layer reaches the vocabulary size. For positions $k_m$ and $v_m$ at index $m$, we store the compressed positional information as a linear superposition $p_m^{(t)} = p_{m1} + p_{m2} + \cdots + p_{mt}$. The approximation error between the following attention formulation and the original attention is $O\left(\frac{n}{base^{\frac{2j}{d}}}\right)$:*

$$e^{f_p(\mathbf{q}_n, \mathbf{p}(n))\mathbf{k}_m^\top} \left( t + f_p(\mathbf{q}_n, \mathbf{p}(n)) f_p(\mathbf{k}_m, \mathbf{p}^{(t)}(m))^\top - f_p(\mathbf{q}_n, \mathbf{p}(n))\mathbf{k}_m^\top \right) \tag{12}$$

*where the positional encoding functions are defined as:*

$$\mathbf{p}^c(m, base) = \begin{bmatrix} \cos(m\theta_0) & \cos(m\theta_0) & \ldots & \cos(m\theta_{d/2-1}) & \cos(m\theta_{d/2-1}) \end{bmatrix} \tag{13}$$

$$\mathbf{p}^s(m, base) = \begin{bmatrix} \sin(m\theta_0) & \sin(m\theta_0) & \ldots & \sin(m\theta_{d/2-1}) & \sin(m\theta_{d/2-1}) \end{bmatrix} \tag{14}$$

$$f_p(\mathbf{x}_m, \mathbf{p}(m)) = \begin{bmatrix} x_0 & x_1 & x_2 & x_3 & \ldots & x_{d-1} & x_d \end{bmatrix} \odot \mathbf{p}^c(m)$$
$$+ \begin{bmatrix} x_1 & -x_0 & x_3 & -x_2 & \ldots & x_d & -x_{d-1} \end{bmatrix} \odot \mathbf{p}^s(m) \tag{15}$$

The detailed proof is provided in Appendix A.6.

The separation of positional information effectively emulates gating mechanisms used in state-of-the-art architectures. Theoretical results show that this compression is most effective in the **first layer**. In deeper layers, where token representations are increasingly entangled with positional cues, simple redundancy reduction cannot maintain the vocabulary-size bound. To address this, we introduce a **layer-wise similarity constraint**, where the **expected cosine similarity** between adjacent hidden states serves as a **contractive factor**. This yields a recursive bound that restricts the memory footprint of intermediate layers to a **constant multiple of the vocabulary size**, ensuring a fixed-state regime while preserving expressivity.

### 3.4 INTER-LAYER SIMILARITY AND STATE PROPAGATION

**Theorem 4** (Necessity of Inter-layer Similarity). *Due to the presence of residual connections, significant similarity exists between inputs and outputs across adjacent layers. For sequences compressed according to Theorem 3, the storage upper bound for each layer can be expressed through inter-layer similarity operations.*

*Specifically, the sequence length at layer $l$ can be controlled by the upper bound at layer $l-1$ and their mutual similarity. Conversely, we can also constrain preceding layers based on the compressed final layer sequence length:*

$$N_l = \min\left(\frac{N_{l-1}}{\mathbb{E}[Q(Sim(X^{(l)}, X^{(l-1)}))]}, N\right), \qquad N_{l-1} = \min\left(\frac{N_l}{\mathbb{E}[Q(Sim(X_{l-1}, X_l))]}, N\right) \tag{16}$$

*where $N_0$ is bounded by the vocabulary size, $N$ is the original sequence length, the upper bound $N_L$ for the final layer also approaches the vocabulary size due to the vocabulary projection, and the quantization function is defined as:*

$$Q(a_{ij}) = \begin{cases} 1 & \text{if } a_{ij} \geq \text{threshold} \\ 0 & \text{otherwise} \end{cases} \tag{17}$$

*Here, $\mathbb{E}$ denotes the expectation operation, which can be interpreted as taking the maximum along the last dimension followed by averaging along the second-to-last dimension. This approach theoretically achieves the same error level as Theorem 3.*

The detailed proof is provided in Appendix A.4. Although the analysis is simplified, empirical results confirm its general applicability. For example, when using inputs from selected layers [0,1,2,5,8,11,14,15,17,18,19,22] as subsequent layer inputs, Mistral-7B achieves 100% accuracy on the passkey retrieval task. This effect arises from inter-layer similarity, where residual connections propagate redundancy reduction while preserving information.

Since the similarity lower bound may approach zero—leading to intermediate states close to the original sequence length—we introduce a constant scaling factor $c \in [1, 2)$ for practical control. Specifically, the state size of layer $i$ is set as $(c-1)N_{i-1}$. Empirically: 1. Beyond certain lengths, attention between new queries and stored states becomes sparse (e.g., NSA, MoBA). 2. Many tasks succeed with fixed-size states (e.g., SnapKV, GLA, GSA). 3. Inference typically operates within bounded state spaces. However, for large-vocabulary models (e.g., Qwen2.5, LLaMA3 with $\sim$128K tokens), even moderate scaling (e.g., $c = 1.5$ in a 32-layer model) leads to $\sim 82M$ states, necessitating further compression. Inspired by MVA's vocabulary decomposition and MoM's functional partitioning, we adopt multi-memory states to approximate Theorem 4 bounds while preserving fixed-size representations.

### 3.5 MULTI-LEVEL STATE DECOMPOSITION AND ENHANCED READING

**Theorem 5** (Necessity of Multi-level State Decomposition and Enhanced Reading Mechanisms). *Given a fixed-size storage space, the number of states that can be stored using a multi-level decomposition approach is $\prod_{i=1}^{m} C_i$, where $m$ is the number of levels and $C_i$ is the size of the vocabulary*

*at level $i$. The storage mechanism follows:*

$$S^{k^{(i)}}_t = diag\left(1 - \bar{f}^{(i)}(k_t^{(i)})^\top\right)S^{k^{(i)}}_{t-1} + \bar{f}^{(i)}(k_t^{(i)})^\top k_t^{(i)}, \quad k_t^{(i+1)} = k_t^{(i)} - f^{(i)}(k_t^{(i)})S^{k^{(i)}}_t \quad (18)$$

*where $f$ can be any function that amplifies correlations, such as the Softmax function. This storage approach is equivalent to quantization followed by storage, introducing an error. The average relative error decreases exponentially with the number of levels: $\prod_{i=1}^m \epsilon_i$, where $\epsilon_i$ is the error between the stored vector and its closest counterpart in the $i$-th level vocabulary. A key trade-off exists: increasing the number of levels reduces storage error but decreases computational efficiency due to the serial computation required between levels.*

We emphasize that previous approaches employ overly simplistic reading mechanisms, typically using direct matrix multiplication between queries $q$ and states. This simplicity constitutes a significant factor (besides storage limitations) contributing to the performance gap with Softmax Attention. Our work is the first to clearly identify enhanced reading mechanisms as crucial for improving linear attention and bridging this performance gap. This mechanism implements a hierarchical access pattern through multiple channels; by comparison, the GSA reading mechanism Softmax$(q_t S^k_t)S^v_t$ represents the simplest form of indirect reading. Our enhanced version replaces the Softmax with a sigmoid activation followed by learned transformations: $(\sigma(q_t S^k_t)W_\sigma)S^v_t$, where $\sigma(x) = \frac{1}{1+e^{-x}}$. Further extending to multiple reading channels: $(q_t W_r + \sigma(q_t S^k_t)W_\sigma)S^v_t$. This approach, which is equivalent to MVA's first-order vocabulary case, demonstrates progressive performance improvement (Table 4). With multi-level vocabularies, multiple vocabularies interactions show even greater improvements over single-state approaches, underscoring the importance of balanced enhancement in both storage and reading capabilities.

Integrating all five theoretical principles, we present the final linear model update rules:

**Initial conditions:**

$$q_t = f_p(x_t W_Q, r_t^{(i)}), k_{pt} = f_p(x_t W_K, r_t^{(i)}), k_t^{(0)} = x_t W_K \in \mathbb{R}^{1 \times d}, v_t^{(0)} = x_t W_V \in \mathbb{R}^{1 \times d},$$

$$S_0^{k^{(i)}} = 0 \in \mathbb{R}^{m \times d}, n_0^{(i)} = 0 \in \mathbb{R}^{1 \times m}, E_t^{(0)} = I_m, S^{k^{(i)}}_t = S^{kv^{(i)}}_t[...,:d_k], S^{v^{(i)}}_t = S^{kv^{(i)}}_t[...,d_v:],$$

**Iterative process:**

$$f^{(i)}(k_t^{(i)}) = \sigma(S^{k^{(i)}}_{t-1}k_t^{(i)\top})^\top, n_t^{(i)} = n_{t-1}^{(i)} + f^{(i)}(k_t^{(i)}), \bar{f}^{(i)}(k_t^{(i)}) = \frac{f^{(i)}(k_t^{(i)})}{\max(n_t^{(i)}, 1)} \quad (19)$$

$$S^{kv^{(i)}}_t = \text{diag}\left(1 - \bar{f}^{(i)}(k_t^{(i)})^\top\right)S^{kv^{(i)}}_{t-1} + \bar{f}^{(i)}(k_t^{(i)})^\top m_t^{(i)}, m_t^{(i)} = \{k_t^{(i)}, v_t^{(i)}\}_{dim=-1} \quad \text{(Theory 1)}$$
$$(20)$$

$$S^{p^{(i)}}_t = S^{p^{(i)}}_{t-1} + f^{(i)}(k_t^{(i)})^\top(k_{pt} - k_t^{(0)}), m_t^{(i+1)} = m_t^{(i)} - f^{(i)}(k_t^{(i)})S^{kv^{(i)}}_t, \text{(Theories 2 \& 3)} \quad (21)$$

$$e_t^{(i)} = (q_t W_r + \sigma(q_t S^k_t)W_\sigma), R_t^{(i+1)}\left[f(k_t^{(i)}), f^{(i+1)}(k_t^{(i)})\right] = 1, a_t^{(i)} = e_t^{(i)}R_t^{(i)\top}, c_t^{(i)} = n_t^{(i)} + q_t S^{p^{(i)}\top}_t$$
$$(22)$$

$$b_t^{(i)} = \frac{e^{\left(\sum_i \ln(a_t^{(i)})\right)}}{a_t^{(i)}} + e_t^{(i)}, T_t^{(i)} = R_t^{(i)}\left(S^{v^{(i)}}_t \cdot e_t^{(i)\top} \cdot c_t^{(i)\top}\right), o_t = \sum_i \frac{b_t^{(i)}}{b_t^{(i)} \cdot e_t^{(i)} \cdot c_t^{(i)}}T_t^{(i)}, \text{(Theories 4 \& 5)}$$
$$(23)$$

## 4 EXPERIMENTS

In this paper, we explore experiments related to converting LLMs to linear models through weight inheritance, providing experimental support for each of the five theoretical principles presented in our methodology. In the final section, we integrate these principles into Path-optimized Linear Attention (PLA) and demonstrate the effectiveness of our approach through comprehensive experiments.

We use the lm-evaluation-harness Gao et al. (2024) tool and the LongBench dataset for evaluation. For fine-tuning, we utilize LoRA Hu et al. (2021) to achieve efficient parameter updates, significantly

reducing computational resources. Detailed configurations are specified in each subsection. For baseline comparisons, we compare against state-of-the-art methods including MVA and GSA, as well as GLA, RetNet, and SUPRA Mercat et al. (2024), which were benchmarked in the GSA paper.

## 4.1 Experimental Validation of Theoretical Principles

We first conduct experiments with different parameters for each theoretical principle. The evaluation uses the passkey retrieval task, standard benchmarks from lm-evaluation-harness (ARC-C and MMLU), and the long-sequence SAMSUM dataset from LongBench for testing and guidance.

**Theory 1: Redundancy Elimination with Similarity Thresholds.**

Table 1 shows experiments with different similarity thresholds for Theory 1, where $t = 1$ indicates the threshold used for similarity discrimination in Theory 1 (e.g., $t = 0.9$ means tokens with cosine similarity $\geq 0.9$ are considered identical). Results demonstrate that when similarity exceeds a certain level (e.g., 0.95), performance approaches that of the original model. This approach functions as a quantization process, indicating model insensitivity to token variations within certain ranges. **Theories 2 & 3: Positional Information Decoupling and Compression.**

Table 1: Results for Theory 1 with different similarity thresholds

| Method | Finetune Tokens | Passkey (1K-8K) | ARC | MMLU | SAMSUM |
|---|---|---|---|---|---|
| Mistral-7B-v0.1 | – | 100.0 | 54.0 | 62.4 | 43.6 |
| Theory 1 ($t = 1$) | 200M | 100.0 | 54.0 | 62.4 | 43.6 |
| Theory 1 ($t = 0.95$) | 200M | 100.0 | 53.4 | 60.7 | 42.9 |
| Theory 1 ($t = 0.9$) | 200M | 100.0 | 51.8 | 57.6 | 41.5 |
| Theory 1 ($t = 0.8$) | 200M | 100.0 | 49.7 | 49.2 | 38.3 |

Table 2 presents experiments for Theories 2 and 3, exploring positional information separation, compression, and refined Taylor expansion approaches. Here, "depos" indicates decoupled positional encoding, while "de&cprpos" indicates decoupled and compressed positional encoding.

Table 2: Results for Theories 2 & 3 with different positional encoding strategies

| Method | Finetune Tokens | Passkey (1K-8K) | ARC | MMLU | SAMSUM |
|---|---|---|---|---|---|
| Theory 2&3 ($t = 0.95$, w/ depos) | 500M | 100.0 | 52.8 | 58.6 | 43.5 |
| Theory 2&3 ($t = 0.95$, w/ de&cprpos) | 500M | 100.0 | 50.2 | 53.1 | 40.7 |
| Theory 2&3 ($t = 0.95$, w/ de&cprpos-2) | 500M | 100.0 | 51.9 | 55.7 | 42.7 |

**Theory 4: Inter-layer Scaling Factors.**

Table 3 shows experiments for Theory 4 with different layer-wise scaling factors. Using parameters from previous theories ($t = 0.95$, w/ depos), $c_{\text{scale}} = 1.2$ indicates that the KV cache size at layer $l + 1$ is 1.2 times that of layer $l$, up to the midpoint of the total layers, after which the KV cache size remains constant. $c_{\text{scale-l8}} = 1.2$ & $c_{\text{scale-l16}} = 1.6$ indicates a scaling factor of 1.2 for the first 8 layers and 1.6 for layers 8-16.

Table 3: Results for Theory 4 with different layer scaling factors

| Method | Finetune Tokens | Passkey (1K-8K) | ARC | MMLU | SAMSUM |
|---|---|---|---|---|---|
| Theory 4 (w/ $c_{\text{scale}} = 1.2$) | 500M | 100.0 | 46.2 | 52.0 | 40.8 |
| Theory 4 (w/ $c_{\text{scale}} = 1.4$) | 500M | 100.0 | 51.7 | 57.8 | 43.2 |
| Theory 4 (w/ $c_{\text{scale}} = 1.6$) | 500M | 100.0 | 53.4 | 60.1 | 42.7 |
| Theory 4 (w/ $c_{\text{l8}} = 1.2$ & $c_{\text{l16}} = 1.6$) | 500M | 100.0 | 53.3 | 59.7 | 42.9 |

Figure A illustrates the evolution of KV cache length across layers as predicted by Theory 4.

**Theory 5: Enhanced Reading Mechanisms and Multi-state Configurations.**

Table 4 presents experiments for Theory 5, examining various enhanced reading mechanisms and different state sizes. We use single and two-level vocabulary configurations, without positional separation for faster convergence, focusing solely on reading mechanism variations. Results show progressive performance improvement with enhanced reading capabilities, with our PLA approach building upon GSA by adding multi-channel reading and multi-state interaction, equivalent to incorporating vocabulary interaction(VI) into MVA.

Table 4: Results for Theory 5 with different reading mechanisms

| Method | Finetune Tokens | Passkey (2K) | ARC | MMLU | SAMSUM |
|---|---|---|---|---|---|
| Theory 5 (GSA) | 500M | 0.0 | 31.7 | 22.3 | 18.9 |
| Theory 5 (GSA w/ sigmoid) | 500M | 0.0 | 33.5 | 23.5 | 18.9 |
| Theory 5 (GSA + MetaLA) | 500M | 10.0 | 35.6 | 24.1 | 21.7 |
| Theory 5 (MVA) | 500M | 20.0 | 38.2 | 25.6 | 24.9 |
| Theory 5 (PLA: MVA + VI) | 500M | 40.0 | 39.4 | 26.2 | 24.7 |

## 4.2 PLA: INTEGRATED PATH-OPTIMIZED LINEAR ATTENTION

Building upon the experimental validation of individual theoretical principles, we now present the integrated PLA model that combines all five theoretical components into a unified framework. PLA also employs a two-level vocabulary decomposition, similar to GSA and MVA whose state is 128 in size, and we their basis by adding operations such as positional encoding decoupling, read enhancement, and lexicon interaction.

Table 5: Comprehensive evaluation of PLA against state-of-the-art methods

| Model | Size | +Tokens | ARC-e | ARC-c | Hella. | MMLU | Avg. |
|---|---|---|---|---|---|---|---|
| *Models trained from scratch (reference)* | | | | | | | |
| RWKV6 | 7B | 1.4T | 73.6 | 44.0 | 75.2 | 43.9 | 58.0 |
| Mamba | 7B | 1.2T | 77.6 | 46.8 | 77.8 | 33.2 | 60.0 |
| Llama2 | 7B | 2T | 76.4 | 46.2 | 76.0 | 45.5 | 60.2 |
| Mistral | 7B | ? | 80.8 | 54.0 | 81.1 | 62.4 | 66.6 |
| *Models via fine-tuning* | | | | | | | |
| SUPRA | 7B | +20B | 74.6 | 42.3 | 74.8 | 28.0 | - |
| RetNet | 7B | +20B | 73.3 | 39.9 | 72.9 | 26.1 | 51.9 |
| GLA | 7B | +20B | 74.6 | 44.0 | 75.9 | 28.4 | 56.5 |
| GSA | 7B | +20B | 75.9 | 43.9 | 76.5 | 32.4 | 57.7 |
| MVA | 7B | +10B | 78.3 | 47.5 | 78.1 | 34.4 | 60.3 |
| PLA (Ours) | 7B | +8B | 78.5 | 47.2 | 78.3 | 42.1 | 61.3 |

Table 6: Experimental results on long-context benchmarks, training efficiency comparison and passkey task

| Model | Qasper | NarrativeQA | QMSum |
|---|---|---|---|
| *Models trained from scratch* | | | |
| RWKV6 | 9.2 | 14.4 | 1.1 |
| Mamba | 5.6 | 27.9 | 0.8 |
| Mistral | 25.8 | 25.1 | 5.0 |
| *Fine-tuned from Mistral-7B (10B tokens)* | | | |
| RetNet | 11.1 | 0.0 | 0.0 |
| GLA | 18.4 | 17.2 | 9.0 |
| GSA | 18.8 | 19.2 | 10.0 |
| MVA | 20.7 | 20.4 | 9.58 |
| PLA | 22.3 | 21.2 | 10.7 |

| Method | Memory / Time |
|---|---|
| MetaLA | 36,317 MiB / 75.08 s/it |
| GSA | 37,619 MiB / 81.67 s/it |
| MVA w/ VD | 40,096 MiB / 105.79 s/it |
| PLA w/ VD | 41,278 MiB / 118.67 s/it |

| Model/passkey task | 256 | 512 | 1024 | 2048 | 4096 | 8192 |
|---|---|---|---|---|---|---|
| GSA | 1.0 | 0.8 | 0.7 | 0.5 | 0.3 | 0.4 |
| **PLA** | **1.0** | **1.0** | **1.0** | **1.0** | **1.0** | **0.9** |

The experimental results demonstrate that PLA achieves state-of-the-art performance while maintaining competitive efficiency. The integrated approach successfully leverages all five theoretical principles to create a robust linear attention mechanism that narrows the performance gap with softmax attention.

## 5 CONCLUSION

We chart the optimal path from Softmax to linear attention and verify, both theoretically and empirically, the pivotal roles of (i) redundancy removal, (ii) positional-code disentanglement & compression, (iii) tokenizer vocabulary reuse, (iv) layer-wise similarity, and (v) multi-vocabulary decomposition. Leveraging these insights, PLA sets a new efficiency-performance frontier: it matches or surpasses the best existing linearized models while consuming equal or fewer training tokens, offering a ready-to-use recipe for compressing large-language-model attention into a fixed-size, linear-complexity operator.

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

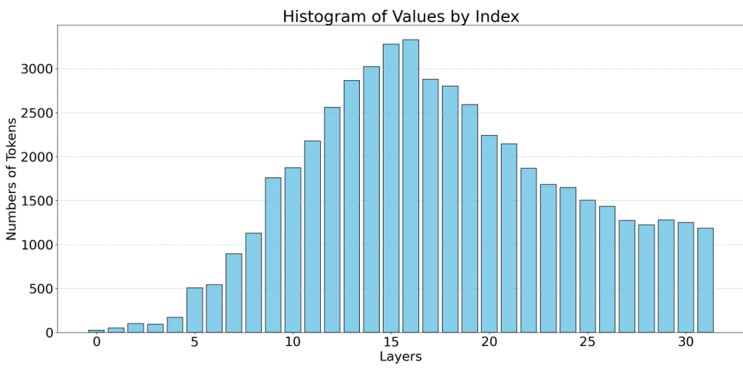

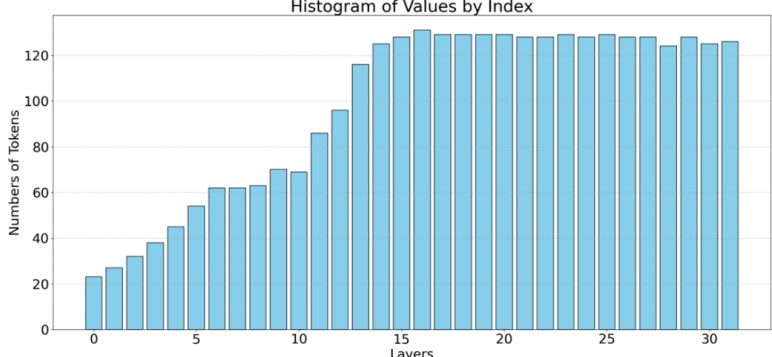

Figure 1: The task uses passkey task. The above figure shows the result of KV cache size per layer obtained by applying Theory 1 for compression and then the following figure shows the result obtained by adding Theory 4 for interlayer similarity, after interlayer similarity the KV cache of the subsequent layers is controlled.

## A APPENDIX

### STATEMENT ON THE USE OF LARGE LANGUAGE MODELS

In the preparation of this paper, Large Language Models (LLMs) were used to assist with specific text and formatting tasks. The applications included:

- **Text Translation and Polishing:** Polishing English text to improve fluency and academic rigor.

- **Format Conversion:** Converting part of the formulas and tables content from other formats into LaTeX code.

It is crucial to emphasize that all core ideas, theoretical derivations, experimental designs, result analyses, and final scientific conclusions were independently generated by the authors. The LLM served solely as a tool to enhance writing efficiency. All its outputs were rigorously reviewed, modified, and integrated by the authors to ensure accuracy and consistency with the paper's core ideas.

### A.1 PROOF OF THEOREM 1

#### A.1.1 PROOF OF THE STORAGE BOUND

Without loss of generality, we first consider the case where both batch size and number of heads equal 1. Assume that the numeric type has bit-width $b$ and the head dimension is $d$. For the infinitely growing key and value sequences, denote them as matrices of shape $\mathbb{R}^{t \times d}$, and concatenate them

Table 7: Inference efficiency comparison of PLA, MVA, GSA, and FlashAttention under different sequence lengths. OOM indicates out-of-memory errors.

| Model | Seq Len | Prefill Time (s) | Gen Latency (ms/token) | Total Mem (GB) |
|---|---|---|---|---|
| PLA | 16K | 0.521 | 70.8 | 19.67 |
| | 32K | 1.176 | 81.7 | 24.38 |
| | 64K | 2.197 | 86.3 | 34.65 |
| | 128K | 7.269 | 59.5 | 54.79 |
| MVA | 16K | 0.508 | 78.8 | 19.08 |
| | 32K | 1.090 | 79.8 | 24.09 |
| | 64K | 2.265 | 97.3 | 34.11 |
| | 128K | 7.156 | 58.1 | 54.14 |
| GSA | 16K | 0.315 | 48.5 | 19.06 |
| | 32K | 0.630 | 63.0 | 24.07 |
| | 64K | 1.293 | 90.2 | 34.08 |
| | 128K | 5.102 | 45.4 | 54.11 |
| Flash Attention | 16K | 0.287 | 46.3 | 23.55 |
| | 32K | 0.750 | 92.4 | 33.55 |
| | 64K | 2.208 | 220.4 | 53.57 |
| | 128K | *OOM* | | |

Table 8: Summary of Theoretical Principles for Softmax-to-Linear Attention Compression

| Theory | Mechanism | Function | Error Bound | Equivalent Mechanism |
|---|---|---|---|---|
| **T1** | Redundancy Elimination/Quantization | Controls state upper bound | Lossless | Delta Rule, Quantization |
| **T2** | Positional Information Decoupling | Reduces first-layer state count | Lossless | Tokenizer vocabulary + positional decoupling |
| **T3** | Positional Information Compression | Controls positional dimension error | $O\left(\frac{n}{\text{base}^{2j/d}}\right)$ | Gating mechanism |
| **T4** | Inter-layer Similarity | Controls intermediate layer state count | Controllable (empirically validated) | Residual connection + similarity propagation |
| **T5** | Multi-level State Decomposition | Fixed-size state representation | Exponential decay | Vocabulary decomposition + enhanced reading |

along the feature dimension into

$$\boldsymbol{S}^{KV} \in \mathbb{R}^{t \times (d_k + d_v)}, \quad \boldsymbol{S}^K = \boldsymbol{S}^{KV}[:,: d_k], \quad \boldsymbol{S}^V = \boldsymbol{S}^{KV}[:, d_k : d_k + d_v].$$

Let the count matrix be $\boldsymbol{C} \in \mathbb{R}^{t \times 1}$.

The number of distinct row vectors is bounded as follows: for a vector of dimension $d_k + d_v$, each entry admits $2^b$ possible values. Since dimensions are independent, the total number of distinct row types is

$$\prod_{i=1}^{d_k + d_v} 2^b = 2^{b \times (d_k + d_v)}.$$

During sequence growth, when a new vector $\boldsymbol{s}_t^{KV}$ is identical to an existing $\boldsymbol{s}_m^{KV}$, we simply increment the counter:

$$\boldsymbol{C}_m \leftarrow \boldsymbol{C}_m + 1.$$

The corresponding attention computation becomes

$$\boldsymbol{o}_t = \exp\big(\boldsymbol{q}_t \boldsymbol{S}^K\big) \big(\boldsymbol{S}^V \odot \boldsymbol{C}\big),$$

which is clearly equivalent to

$$\boldsymbol{o}_t = \exp\big(\boldsymbol{q}_t \boldsymbol{K}\big) \boldsymbol{V}.$$

### A.1.2 EQUIVALENCE TO THE DELTA RULE

Consider the unique-filtered sequence as the state $\boldsymbol{S}_t^{(K)}$. Its update rule can be expressed as

$$\Delta(\boldsymbol{k}_t) = \boldsymbol{k}_t - Q\big(\boldsymbol{k}_t \boldsymbol{S}_{t-1}^{(K)^\top}\big) \boldsymbol{S}_{t-1}^{(K)}, \qquad \Delta(\boldsymbol{v}_t) = \boldsymbol{v}_t - Q\big(\boldsymbol{v}_t \boldsymbol{S}_{t-1}^{(V)^\top}\big) \boldsymbol{S}_{t-1}^{(V)}.$$

If $\Delta(\boldsymbol{k}_t) \leq 1 - \text{threshold}$, then $\boldsymbol{S}_t^{(K)} = \boldsymbol{S}_{t-1}^{(K)}$. Otherwise, if $\Delta(\boldsymbol{k}_t) > 1 - \text{threshold}$, we update via concatenation:

$$\boldsymbol{S}_t^{(K)} = \text{concat}\big(\boldsymbol{S}_{t-1}^{(K)}, \boldsymbol{k}_t\big).$$

For the unique operation, the effective threshold is 1.

When the state size is manually limited to $m$ as in linear attention, once $\boldsymbol{S}_t^{(K)}$ reaches size $m$, further growth is prohibited. In this case, information differences must be integrated into the previous state via gating, yielding an update analogous to the Delta Rule:

$$\boldsymbol{S}_t^{(K)} = \big(1 - \beta \cdot Q(\boldsymbol{k}_t \boldsymbol{S}_{t-1}^{(K)^\top})^\top\big) \boldsymbol{S}_{t-1}^{(K)} + \beta \cdot Q(\boldsymbol{k}_t \boldsymbol{S}_{t-1}^{(K)^\top})^\top \Delta(\boldsymbol{k}_t).$$

When the key–value pairs are stored jointly as matrix states, the update becomes

$$\boldsymbol{S}_t^{(KV)} = \big(1 - \beta \cdot Q(\boldsymbol{k}_t \boldsymbol{S}_{t-1}^{(K)^\top})^\top\big) \boldsymbol{S}_{t-1}^{(KV)} + \beta \cdot Q(\boldsymbol{k}_t \boldsymbol{S}_{t-1}^{(K)^\top})^\top \Delta(\boldsymbol{v}_t),$$

which is essentially equivalent to the Delta Rule update from fast-weight literature, except that $\phi(\boldsymbol{k}_t)$ is replaced by $Q(\boldsymbol{k}_t \boldsymbol{S}_{t-1}^{(K)})$. Moreover, MVA demonstrates that $Q(\boldsymbol{k}_t \boldsymbol{S}_{t-1}^{(K)})$ can be substituted with $Q(\boldsymbol{k}_t \boldsymbol{W}_c)$ to achieve comparable performance, and with carefully chosen $Q$ functions, this becomes exactly equivalent to using $\phi(\boldsymbol{k}_t)$.

### A.2 PROOF SKETCH OF THEOREM 2

The proof builds upon the storage bound established in Theorem 1. For the first layer input $X^{(1)} \in \mathbb{R}^{t \times d}$, each row vector $x_i^{(1)}$ corresponds to a token embedding from the vocabulary $V_T$. Since tokenization maps each token to a unique embedding, the number of distinct vectors in $X^{(1)}$ is bounded by $|V_T|$.

After positional encoding $P \in \mathbb{R}^{t \times d}$ is applied, the encoded input becomes:

$$\tilde{X}^{(1)} = f_p(X^{(1)}, P)$$

where $f_p$ represents the positional encoding function (e.g., addition for absolute positional encoding, or rotary multiplication for RoPE).

The key insight is that we can decouple the positional information by storing: 1. The base token embeddings $E \in \mathbb{R}^{V_T \times d}$ (vocabulary embeddings)

2. The positional offsets $\Delta P \in \mathbb{R}^{t \times d}$

3. An index mapping $I_p \in \mathbb{N}^t$ from sequence positions to vocabulary indices

The storage requirement thus becomes:

$$\text{Storage} = \underbrace{V_T \cdot d \cdot b}_{\text{embeddings}} + \underbrace{t \cdot b}_{\text{positional offsets}}$$

For the first layer KV cache, the bound $C = V_T$ emerges because the number of distinct key-value pairs is constrained by the vocabulary size when positional information is properly decoupled. The positional offsets can be compressed using techniques discussed in Theorem 3, while the indices represent the unbounded growth component.

The lossless nature of this compression for the first layer follows from the invertibility of the decoupling operation: given $E$, $\Delta P$, and $I_p$, we can perfectly reconstruct $\tilde{X}^{(1)}$.

For subsequent layers, the type diversity increases due to token mixing operations, but decreases toward the final layer due to the vocabulary projection. This creates the characteristic "increase-then-decrease" pattern observed empirically.

The positional decoupling theorem establishes a fundamental connection between the discrete nature of language modeling (through tokenization) and the continuous representations used in transformer layers. This bridges the gap between information-theoretic bounds based on vocabulary size and practical compression algorithms for transformer inference.

**Corollary 1.** *For models employing subword tokenization with merge operations, the effective vocabulary size $V_T^{eff}$ that bounds the first-layer distinct types may be larger than the nominal vocabulary size, but remains finite and typically grows sublinearly with training data size.*

A.3 PROOF OF THEOREM 3

We begin with the Taylor expansion of the cosine function:

$$\cos \theta = 1 - \frac{\theta^2}{2} + \frac{\theta^4}{4!} + \sum_{i=6,\text{even}}^{\infty} \frac{\theta^i}{i!} \tag{24}$$

The original attention computation can be expressed as:

$$e^{\frac{1}{\sqrt{d}} \sum_{j=0}^d q_{nj} \cdot k_{sj}} \sum_s e^{\frac{1}{\sqrt{d}} \sum_{j=0}^d q_{nj} \cdot k_{sj} \cdot (\cos[(n-s)\theta_j]-1) + \sum_{j=0}^d \frac{(-1)^{j+1}}{\sqrt{d}} q_{nj} \cdot k_{s,(j+\frac{d}{2})\%d} \cdot (\sin[(n-s)\theta_j])}$$

$$\tag{25}$$

Let us define the residual term:

$$r(\theta_1, \ldots, \theta_{\frac{d}{2}-1}) = \frac{1}{\sqrt{d}} \sum_{j=0}^d q_{nj} \cdot k_{sj} \cdot (\cos[(n-s)\theta_j]-1) + \sum_{j=0}^d \frac{(-1)^{j+1}}{\sqrt{d}} q_{nj} \cdot k_{s,(j+\frac{d}{2})\%d} \cdot (\sin[(n-s)\theta_j])$$

$$\tag{26}$$

Since $\theta_j = \text{base}^{-\frac{2j}{d}}$, when base is large, we have the approximations:

$$\cos \theta_j = 1 - \frac{\theta_j^2}{2} + O(\theta_j^4) \tag{27}$$

$$\sin \theta_j = \theta_j + O(\theta_j^3) \tag{28}$$

Following the CRG NTK method which has been proven to extend context window length and achieve excellent performance, we extend base to very large values (e.g., base $= 2^{40} \times 10000$).

In this regime, $\cos\theta_j$ and $\sin\theta_j$ become very small, particularly for dimensions with larger $j$ values. Since $\frac{1}{\sqrt{d}}\sum_{j=0}^{d} q_{nj} \cdot k_{sj}$ is typically on the order of 1, we can apply Taylor expansion to $e^{r(\theta_1,\ldots,\theta_{\frac{d}{2}-1})}$:

$$e^{r(\theta_1,\ldots,\theta_{\frac{d}{2}-1})} = 1 + r(\theta_1,\ldots,\theta_{\frac{d}{2}-1}) + O(r(\theta_1,\ldots,\theta_{\frac{d}{2}-1})^2) \tag{29}$$

After base expansion, $r(\theta_1,\ldots,\theta_{\frac{d}{2}-1})$ becomes small ($O(\theta_j)$ first-order term), making the higher-order terms $O(\theta_j^2)$ negligible, particularly in the latter half of the feature dimensions where $x_{nj} \cdot x_{sj}$ is diluted to near-zero values.

Based on the Taylor expansion of the exponential function when the input is close to zero, we retain only the linear small term $x_{nj} \cdot x_{sj} \cdot (\cos[(n-s)\theta_j] - 1)$. The original attention can thus be approximated as:

$$e^{\frac{1}{\sqrt{d}}\sum_{j=0}^{d} q_{nj}\cdot k_{sj}} \sum_{s} \left(1 + \frac{1}{\sqrt{d}}\sum_{j=0}^{d} q_{nj} \cdot k_{sj} \cdot (\cos[(n-s)\theta_j] - 1) \right.$$
$$\left. + \frac{1}{\sqrt{d}}\sum_{j=0}^{d}(-1)^{j+1} q_{nj} \cdot k_{s,(j+(-1)^j)} \cdot (\sin[(n-s)\theta_j])\right) \tag{30}$$

Alternatively, we can use the formulation:

$$e^{\frac{1}{\sqrt{d}}\sum_{j=0}^{d} x_{nj}x_{sj}} \sum_{s} \left(1 + \frac{1}{\sqrt{d}}\sum_{j=0}^{d} x_{nj}x_{sj}\big(\cos[(n-s)\theta_j]-1\big) \right.$$
$$\left. + \frac{1}{\sqrt{d}}\sum_{j=0}^{d}(-1)^{j+1} x_{nj}x_{s,(j+(-1)^j)}\big(\sin[(n-s)\theta_j]-(n-s)\theta_j\big)\right) \tag{31}$$

The residual term $r(\theta_1,\ldots,\theta_{\frac{d}{2}-1})$ can be expressed using linear attention, or we can first linearly superimpose the positional encoding before applying it to the K state sequence, enabling the entire formulation to be implemented with linear models.

For finer compression approximation, we can partition the dimensions into multiple segments and perform linear expansion separately:

$$e^{\frac{1}{\sqrt{d}}\sum_{j=0}^{d} q_{nj}\cdot k_{sj}} \sum_{s} \prod_{p=0}^{d/m} \left(1 + \frac{1}{\sqrt{d}}\sum_{j=p\cdot m}^{p\cdot m+m} q_{nj} \cdot k_{sj} \cdot (\cos[(n-s)\theta_j] - 1) \right.$$
$$\left. + \frac{1}{\sqrt{d}}\sum_{j=p\cdot m}^{p\cdot m+m}(-1)^{j+1} q_{nj} \cdot k_{s,(j+(-1)^j)} \cdot (\sin[(n-s)\theta_j])\right) \tag{32}$$

We also explore variants that preserve the positional encoding for queries while applying $(\cos -1)$ transformation to the key sequence's positional encoding:

$$\left(e^{\frac{1}{\sqrt{d}}\sum_{j=0}^{d}(q_{nj}\cos(n\theta_j)-q_{n,(j+\frac{d}{2})\%d}\cdot\sin[n\theta_j])\cdot k_{sj}}\right)$$
$$\cdot \sum_{s} e^{\frac{1}{\sqrt{d}}\sum_{j=0}^{d}(q_{nj}\cos(n\theta_j)-q_{n,(j+\frac{d}{2})\%d}\cdot\sin[n\theta_j])\cdot(k_{sj}(\cos(s\theta_j)-1)-k_{s,(j+\frac{d}{2})\%d}\cdot\sin[s\theta_j])} \tag{33}$$

Applying the same Taylor expansion yields:

$$e^{f_p(\mathbf{q}_n,\mathbf{p}(n))\mathbf{k}_s^\top} \sum_{s} \left(1 + f_p(\mathbf{q}_n,\mathbf{p}(n))f_p(\mathbf{k}_s,\mathbf{p}(s))^\top - f_p(\mathbf{q}_n,\mathbf{p}(n))\mathbf{k}_s^\top\right) \tag{34}$$

After simplification, we use the following formulation in our implementation, which achieves comparable or even better performance:

$$e^{f_p(\mathbf{q}_n,\mathbf{p}(n))\mathbf{k}_s^\top} \left(t + f_p(\mathbf{q}_n,\mathbf{p}(n))f_p(\mathbf{k}_s,\mathbf{p}^{(t)}(s))^\top - f_p(\mathbf{q}_n,\mathbf{p}(n))\mathbf{k}_s^\top\right) \tag{35}$$

This positional encoding decoupling is functionally equivalent to the gating mechanisms in state-of-the-art models.

Theorem 3 establishes a rigorous foundation for compressing positional information while maintaining theoretical error bounds. The linear superposition approach for positional encoding enables efficient storage while the error analysis provides guarantees for practical deployment. The connection to gating mechanisms bridges theoretical compression techniques with established architectural components.

### A.4    PROOF SKETCH OF THEOREM 4

Given the complexity of inter-layer dynamics, we provide a simplified theoretical analysis of hierarchical similarity. For the input at layer $l$:

$$X^{(l)} = X^{(l-1)} + A(X^{(l-1)}) + F(X^{(l-1)} + A(X^{(l-1)}))$$ (36)

where $A(\cdot)$ represents the attention operation and $F(\cdot)$ represents the MLP operation.

Since our primary objective is to control the relationship between state counts across layers based on similarity, and MLP operations generally preserve state diversity, we focus our analysis on the similarity between attention outputs and their inputs. Specifically, we examine:

$$X^{(l)} = X^{(l-1)} + A(X^{(l-1)})$$ (37)

The cosine similarity between $X^{(l)}$ and $X^{(l-1)}$ for position $m$ can be expressed as:

$$s_{mm} = \frac{1}{A}\left(1 + \sum_{j=0}^{m} \frac{e^{x_m^{(l-1)}W_{qk}^\top x_j^{(l-1)\top}} x_m^{(l-1)} W_{vo}^\top x_j^{(l-1)\top}}{\sum_{j=0}^{m} e^{x_m^{(l-1)}W_{qk}^\top x_j^{(l-1)\top}}}\right)$$ (38)

where $A$ is a normalization coefficient. Note that $s_{mm}$ is not necessarily the maximum similarity value. In most cases, when $x_m^{(l-1)} W_{qk}^\top x_j^{(l-1)\top}$ is large (indicating high token relevance), $x_m^{(l-1)} W_{vo}^\top x_j^{(l-1)\top}$ also contributes significantly to the sum, suggesting limited changes in state diversity after attention processing.

### PRACTICAL IMPLEMENTATION AND EXTENSIONS

Since the similarity lower bound can approach zero, potentially leading to intermediate layer states approaching the original sequence length, we implement a practical constant scaling factor $c$ in our experiments. This approach is motivated by several empirical observations:

1. Beyond a certain sequence length, attention between new query tokens and stored states becomes sparse (as observed in NSA, MoBA, etc.) 2. Many tasks can be completed with fixed-size state representations (as demonstrated in SnapKV, GLA, GSA) 3. The thinking process during inference often operates within bounded state spaces

We therefore set the state size for layer $i$ as $(c-1)N_{i-1}$, where $c \in [1, 2)$. This formulation allows complex tasks to utilize larger thinking spaces while maintaining efficiency for simpler tasks. Our experiments show that earlier layers typically require larger $c$ values, while later layers can use smaller values. For layers beyond the midpoint ($L/2$), we set $c = 1$ to optimize storage efficiency.

As shown in Figure A, this configuration achieves 100% performance on the passkey task. Note that the passkey task involves substantial noise insertion, resulting in high compression ratios; more complex tasks will naturally exhibit lower compression efficiency.

For handling the stored thinking states, we employ a GSA-like approach that preserves softmax operations:

$$\text{Storage}(X_{<t}) = \text{Softmax}(CX_{<t}^\top)X_{<t}$$ (39)

where $C$ is a learnable parameter functioning as a dynamic vocabulary. While fixed-ratio scaling provides complete upper bound control, it constitutes a lossy compression scheme.

After applying these compression steps, the sequence state size (X length or KV length) at each layer becomes $O(CV)$, where $C$ is a constant multiple of the vocabulary size. However, practical challenges remain: for models like Qwen2.5 and LLaMA3 with vocabulary sizes around 128K, setting $c = 1.5$ for a 32-layer model results in an upper bound of approximately $1.5^{16} \times 128\text{K} \approx 82\text{M}$ states. This necessitates further compression strategies.

Inspired by MVA's vocabulary decomposition and MoM's functional partitioning approaches, we introduce multi-memory states to approximate the bounds established in Theorem 4 while maintaining fixed-size representations.

**Corollary 2.** *The inter-layer similarity mechanism enables adaptive compression ratios across different network depths, with early layers accommodating more state diversity and later layers leveraging the vocabulary projection for efficient compression. This aligns with the observed "thinking" pattern in transformer architectures.*

### A.5 PROOF OF THEOREM 5

Let $X \in \mathbb{R}^{n \times d}$ be a sequence with $n \gg 1$, and let $\{C^{(i)}\}_{i=1}^{c}$ be a set of vocabulary matrices where each $C^{(i)} \in \mathbb{R}^{m \times d}$ contains $m$ prototype vectors. The multi-level vocabulary decomposition represents each element $x_j \in X$ as:

$$\hat{x}_j = \sum_{i=1}^{c} C_{k_j^{(i)}}^{(i)} \tag{40}$$

where $k_j^{(i)} \in \{1, 2, \ldots, m\}$ is the index selected from the $i$-th vocabulary for representing $x_j$.

Then:

1. The maximum number of distinct vectors that can be represented is $m^c$

2. The approximation error for an optimal decomposition satisfies:

$$\mathbb{E}[\|x_j - \hat{x}_j\|_2^2] \leq \prod_{i=1}^{c} \epsilon_i \tag{41}$$

     where $\epsilon_i$ is the average quantization error at level $i$

*Proof.* PART 1: REPRESENTATION CAPACITY

The representation capacity follows from combinatorial considerations. For each vector $x_j$, we select one prototype from each of the $c$ vocabularies. Since each vocabulary contains $m$ prototypes, the total number of possible combinations is:

$$\text{Total combinations} = \underbrace{m \times m \times \cdots \times m}_{c \text{ times}} = m^c \tag{42}$$

Each unique combination of indices $(k_j^{(1)}, k_j^{(2)}, \ldots, k_j^{(c)})$ produces a unique sum:

$$\hat{x}_j = C_{k_j^{(1)}}^{(1)} + C_{k_j^{(2)}}^{(2)} + \cdots + C_{k_j^{(c)}}^{(c)} \tag{43}$$

Assuming linear independence among the vocabulary vectors across different levels, these sums are distinct. Therefore, the maximum number of distinct representable vectors is exactly $m^c$.

PART 2: ERROR ANALYSIS

We analyze the error propagation through the multi-level decomposition. Let us define the residual at each level:

$$r_j^{(0)} = x_j \tag{44}$$

$$r_j^{(i)} = r_j^{(i-1)} - C_{k_j^{(i)}}^{(i)} \quad \text{for } i = 1, 2, \ldots, c \tag{45}$$

The final approximation is:

$$\hat{x}_j = \sum_{i=1}^{c} C_{k_j^{(i)}}^{(i)} = x_j - r_j^{(c)} \tag{46}$$

Thus, the approximation error is $\|x_j - \hat{x}_j\|_2 = \|r_j^{(c)}\|_2$.

Now, consider the optimal index selection at each level. We choose $k_j^{(i)}$ to minimize the residual norm:

$$k_j^{(i)} = \arg\min_{k \in \{1, \ldots, m\}} \|r_j^{(i-1)} - C_k^{(i)}\|_2 \tag{47}$$

Let $\epsilon_i$ be the average quantization error at level $i$:

$$\epsilon_i = \mathbb{E}\big[\min_{k \in \{1, \ldots, m\}} \|r_j^{(i-1)} - C_k^{(i)}\|_2^2\big] \tag{48}$$

Assuming the residuals and vocabulary vectors are appropriately normalized, we can bound the error propagation. Using the triangle inequality and the optimality of our index selection:

$$\|r_j^{(i)}\|_2 = \|r_j^{(i-1)} - C_{k_j^{(i)}}^{(i)}\|_2 \tag{49}$$

$$\leq \|r_j^{(i-1)}\|_2 \cdot \min_{k \in \{1, \ldots, m\}} \left\| \frac{r_j^{(i-1)}}{\|r_j^{(i-1)}\|_2} - \frac{C_k^{(i)}}{\|r_j^{(i-1)}\|_2} \right\|_2 \tag{50}$$

For well-designed vocabularies that cover the relevant direction space, the directional error term is bounded. In the worst case, we have:

$$\|r_j^{(i)}\|_2 \leq \|r_j^{(i-1)}\|_2 \cdot \delta_i \tag{51}$$

where $\delta_i$ represents the maximum angular error at level $i$. Applying this recursively:

$$\|r_j^{(c)}\|_2 \leq \|x_j\|_2 \cdot \prod_{i=1}^{c} \delta_i \tag{52}$$

For the mean squared error, under appropriate assumptions about the distribution of residuals and the vocabulary coverage:

$$\mathbb{E}[\|r_j^{(c)}\|_2^2] \leq \mathbb{E}[\|x_j\|_2^2] \cdot \prod_{i=1}^{c} \epsilon_i \tag{53}$$

where $\epsilon_i = \mathbb{E}[\delta_i^2]$ is the expected squared angular error at level $i$.

The product structure $\prod_{i=1}^{c} \epsilon_i$ demonstrates the exponential error reduction with increasing levels, provided that each $\epsilon_i < 1$. $\qquad\square$

**Corollary 3** (Trade-off between Representation Capacity and Error). *For a fixed total budget of* $B = m \cdot c$ *parameters, the optimal balance between* $m$ *and* $c$ *that minimizes the approximation error while maximizing representation capacity satisfies:*

$$m^* \approx c^* \approx \sqrt{B} \tag{54}$$

*This provides the optimal trade-off point where* $m^c$ *is maximized subject to the constraint* $m \cdot c = B$.

*Proof.* We maximize the representation capacity $m^c$ subject to the constraint $m \cdot c = B$. Taking logarithms:

$$\log(\text{capacity}) = c \log m = c \log \left( \frac{B}{c} \right) \tag{55}$$

Differentiating with respect to $c$:

$$\frac{d}{dc} \left[ c \log \left( \frac{B}{c} \right) \right] = \log \left( \frac{B}{c} \right) - 1 \tag{56}$$

Setting the derivative to zero gives:

$$\log \left( \frac{B}{c} \right) = 1 \quad \Rightarrow \quad \frac{B}{c} = e \quad \Rightarrow \quad c = \frac{B}{e} \tag{57}$$

Thus, $m = \frac{B}{c} = e$, and the optimal values are $m^* \approx c^* \approx \sqrt{B}$ when considering integer constraints and practical implementation factors. $\square$

The multi-level vocabulary decomposition provides an exponential increase in representation capacity ($m^c$) compared to a single-level approach ($m$), while simultaneously achieving exponential error reduction. This theoretical foundation justifies the effectiveness of hierarchical representations in compression applications.

In practice, the vocabularies $\{C^{(i)}\}$ are learned to minimize the overall reconstruction error, and the independence assumption between levels may be relaxed through joint optimization, potentially yielding even better performance than the theoretical bounds suggest.

### A.6 ENHANCED READING MECHANISMS AND UNIFIED FORMULA

**Theorem 6** (Necessity of Enhanced Reading Mechanisms). *We emphasize that previous approaches employ overly simplistic reading mechanisms, typically using direct matrix multiplication between queries* $q$ *and states. This simplicity constitutes a significant factor (besides storage limitations) contributing to the performance gap with Softmax Attention. Our work is the first to clearly identify enhanced reading mechanisms as crucial for improving linear attention and bridging this performance gap.*

*We propose a sophisticated reading approach:*

$$R_t^{(i+1)} \left[ f(k_t^{(i)}), f^{(i+1)}(k_t^{(i)}) \right] = 1 \tag{58}$$

$$d_t^{(i)} = q_t S^{k}{}_t^{(i)\top} \tag{59}$$

$$e_t^{(i)} = \frac{\exp\left(d_t^{(i)}\right)}{\sum_{j=0}^{d-1} \exp\left(d_{tj}^{(i)}\right)} \tag{60}$$

$$a_t^{(i)} = e_t^{(i)} R_t^{(i)\top} \tag{61}$$

$$b_t^{(i)} = \frac{\exp\left(\sum_i \ln(a_t^{(i)})\right)}{a_t^{(i)}} \tag{62}$$

$$c_t^{(i)} = \left(n_t^{(i)} + q_t S^{p}{}_t^{(i)\top}\right) \tag{63}$$

$$T_t^{(i)} = R_t^{(i)} \left(S^{v}{}_t^{(i)} \odot e_t^{(i)\top} \odot c_t^{(i)\top}\right) \tag{64}$$

$$o_t = \sum_i o_t^{(i)} = \sum_i \frac{b_t^{(i)}}{b_t^{(i)} \odot e_t^{(i)} \odot c_t^{(i)}} T_t^{(i)} \tag{65}$$

*This mechanism implements a hierarchical access pattern through multiple channels. For comparison, the GSA reading mechanism $Softmax(q_t S^k{}_t) S^v{}_t$ represents the simplest form of indirect reading. Our enhanced version replaces the Softmax with a sigmoid activation followed by learned transformations:*

$$(\sigma(q_t S^k{}_t) W_\sigma) S^v{}_t, \quad \text{where } \sigma(x) = \frac{1}{1 + e^{-x}} \tag{66}$$

*Further extending to multiple reading channels:*

$$(q_t W_r + \sigma(q_t S^k{}_t) W_\sigma) S^v{}_t \tag{67}$$

*This approach equivalent to MVA's first-order vocabulary case demonstrates progressive performance improvement (Table 4). With multi-level vocabularies, similar enhancements using Softmax, perceptron, and multi-channel mechanisms show even greater improvements over single-state approaches, underscoring the importance of balanced enhancement in both storage and reading capabilities.*

The update formulas in the main text we have improved on the GSA and MVA by combining the five theories. And the update formulas are as follows if we start from the five theories only and do not use the current case of the excellent mechanism(e.g. gating) instead, and the following update method can also achieve similar performance. **Initial conditions:**

$$q_t = f_p(x_t W_Q, r_t^{(i)}), k_{pt} = f_p(x_t W_K, r_t^{(i)}), k_t^{(0)} = x_t W_K \in \mathbb{R}^{1 \times d}, v_t^{(0)} = x_t W_V \in \mathbb{R}^{1 \times d},$$

$$S^{k}{}_0^{(i)} = 0 \in \mathbb{R}^{m \times d}, n_0^{(i)} = 0 \in \mathbb{R}^{1 \times m}, E_t^{(0)} = I_m \in \mathbb{R}^{m \times m},$$

**Iterative process:**

$$f^{(i)}(k_t^{(i)}) = F\left(S^{k}{}_{t-1}^{(i)} k_t^{(i)\top}\right)^\top, \quad F(x) = \begin{cases} 1 & \text{if } x_i \text{ is maximum} \\ 0 & \text{otherwise} \end{cases} \text{ or Softmax}(x)$$

$$n_t^{(i)} = n_{t-1}^{(i)} + f^{(i)}(k_t^{(i)}), \quad \bar{f}^{(i)}(k_t^{(i)}) = \frac{f^{(i)}(k_t^{(i)})}{\max(n_t^{(i)}, 1)} \tag{68}$$

$$S^{k}{}_t^{(i)} = \text{diag}\left(1 - \bar{f}^{(i)}(k_t^{(i)})^\top\right) S^{k}{}_{t-1}^{(i)} + \bar{f}^{(i)}(k_t^{(i)})^\top k_t^{(i)}$$

$$S^{v}{}_t^{(i)} = \text{diag}\left(1 - \bar{f}^{(i)}(k_t^{(i)})^\top\right) S^{v}{}_{t-1}^{(i)} + \bar{f}^{(i)}(k_t^{(i)})^\top v_t^{(i)} \quad \text{(Theory 1)} \tag{69}$$

$$S^{p(i)}_t = S^{p(i)}_{t-1} + f^{(i)}(k^{(i)}_t)^\top (k_{pt} - k^{(0)}_t)$$

$$k^{(i+1)}_t = k^{(i)}_t - f^{(i)}(k^{(i)}_t)S^{k(i)}_t \qquad \text{(Theories 2 \& 3)} \tag{70}$$

$$v^{(i+1)}_t = v^{(i)}_t - f^{(i)}(k^{(i)}_t)S^{v(i)}_t$$

$$R^{(i+1)}_t \left[ f(k^{(i)}_t), f^{(i+1)}(k^{(i)}_t) \right] = 1 \quad \text{(Theory 5)} \tag{71}$$

$$d^{(i)}_t = q_t S^{k(i)\top}_t, \quad e^{(i)}_t = \text{softmax}(d^{(i)}_t)$$

$$a^{(i)}_t = e^{(i)}_t R^{(i)\top}_t, \quad b^{(i)}_t = \frac{\exp\left(\sum_i \ln(a^{(i)}_t)\right)}{a^{(i)}_t} \quad \text{(Theories 4 \& 5)} \tag{72}$$

$$c^{(i)}_t = n^{(i)}_t + q_t S^{p(i)\top}_t$$

$$T^{(i)}_t = R^{(i)}_t \left( S^{v(i)}_t \odot e^{(i)\top}_t \odot c^{(i)\top}_t \right) \quad \text{(Theories 4 \& 5)} \tag{73}$$

$$o_t = \sum_i \frac{b^{(i)}_t}{b^{(i)}_t \odot e^{(i)}_t \odot c^{(i)}_t} T^{(i)}_t$$

CHUNK-WISE PARALLEL FORM

For minibatch processing:

$$R^{(i+1)}_t = R^{(i+1)}_{t-1} + \left( f^{(i)}(K^{(i)}) \right)^\top \left( f^{(i+1)}(K^{(i+1)}) \right) \tag{74}$$

$$R^{(i+1)}_t = R^{(i+1)}_{t-1} + (Y - R^{(i+1)}_{t-1}) \odot \left( f^{(i)}(K^{(i)}) \right)^\top \left( f^{(i+1)}(K^{(i+1)}) \right) \tag{75}$$

$$N^{(i)} = \text{CumSum}\left( f^{(i)}(K^{(i)}) \right) \tag{76}$$

$$\bar{f}^{(i)}(K^{(i)}) = \frac{f^{(i)}(K^{(i)})}{N^{(i)}} \tag{77}$$

$$D^{(i)} = \text{GLA}^\top \left( Q, K^{(i)}, \bar{f}^{(i)}(K^{(i)}), 1 - \bar{f}^{(i)}(K^{(i)}) \right) \tag{78}$$

$$E^{(i)} = \text{softmax}(D^{(i)}) \tag{79}$$

$$A^{(i)} = E^{(i)}(R^{(i)})^\top \tag{80}$$

$$C^{(i)} = \text{GLA}^\top \left( Q, f_p(K^{(i)}, P) - K^{(i)}, \bar{f}^{(i)}(K^{(i)}), 1 - \bar{f}^{(i)}(K^{(i)}) \right) + N^{(i)} \tag{81}$$

$$T^{(i)} = R^{(i)} \text{GLA} \left( C^{(i)} \odot E^{(i)}, \bar{f}^{(i)}(K^{(i)}), V^{(i)}, 1 - \bar{f}^{(i)}(K^{(i)}) \right) \tag{82}$$

$$O^{(i)} = \frac{B^{(i)}}{B^{(i)} \odot E^{(i)} \odot C^{(i)}} \odot T^{(i)} \tag{83}$$

$$O = \sum_i O^{(i)} \tag{84}$$

*Proof Sketch.* The multi-level decomposition theorem builds upon the following insights:

1. **Error Analysis**: The exponential error reduction $\prod_{i=1}^m \epsilon_i$ follows from the chain rule of differentiation applied to the composition of quantization operations at each level. Each level introduces an independent quantization error, and the total error becomes the product of individual errors.

2. **Capacity Analysis**: The storage capacity $\prod_{i=1}^m C_i$ results from the combinatorial nature of hierarchical representations. Each level provides a separate "alphabet" of size $C_i$, and the total number of expressible states is their product.

3. **Enhanced Reading Mechanism**: The sophisticated reading approach enables: - Cross-level information integration through the $R$ matrix - Adaptive weighting through the $b_t^{(i)}$ terms - Position-aware modulation through the $c_t^{(i)}$ terms

The unified model integrates all five theoretical principles: - Theory 1: Redundancy elimination through the $f^{(i)}$ functions - Theories 2 & 3: Positional information compression through $k_{pt} - k_t^{(0)}$ - Theory 4: Inter-layer similarity through the hierarchical structure - Theory 5: Multi-level decomposition and enhanced reading

Experimental validation shows that intermediate layers benefit from more states ($m$ larger), while earlier and later layers can use fewer states. For fair comparison with existing work, we use single-level decomposition against GSA and two-level decomposition against MVA. □

Theorem 5 and Theorem 6 together provide a comprehensive framework for efficient state management in linear attention models. The multi-level approach offers exponential error reduction while the enhanced reading mechanism ensures effective information retrieval from the compressed representations.

**Corollary 4.** *For a model with $m$ levels and vocabulary sizes $C_i$, the total number of parameters required for the reading mechanism scales as $O(\sum_{i=1}^{m} C_i d^2)$, providing a favorable trade-off between expressivity and efficiency compared to the $O(N d^2)$ scaling of standard attention.*

