# OpenReview forum: "PLA: The Optimal Path from Softmax Attention to Linear Models via KV Cache Compression"
_ICLR.cc/2026/Conference — Submitted to ICLR 2026_

### Official Review · Reviewer_1Eue · 2025-10-27

**Soundness:** 3
**Presentation:** 4
**Contribution:** 2
**Rating:** 4
**Confidence:** 3

**Summary:**

The paper, “PLA: The Optimal Path from Softmax Attention to Linear Models via KV Cache Compression,” aims to establish a theoretically grounded connection between Softmax attention and linear attention. The authors argue that existing linear models (e.g., Performer, GSA, MVA) lack a systematic theoretical bridge to Softmax attention and propose five compression-based theoretical principles that describe how to transform Softmax attention into a linear, fixed-state formulation.

**Strengths:**

1. Provides a systematic theoretical framework connecting Softmax and linear attention through the lens of cache compression.
2. The five derived principles are interpretable and consistent with known empirical behaviors (e.g., redundancy of KV cache, similarity between residual layers).
3. Offers a unified perspective integrating ideas from fast-weight models, memory compression, and linearized attention.

**Weaknesses:**

1. Limited conceptual novelty: The proposed “optimal path” mainly restates and integrates ideas already seen in pruning, cache compression, and linear attention. There is no fundamentally new attention mechanism.
2. Lack of quantitative efficiency evidence: Despite the focus on bounded-state inference, the paper does not report runtime, memory, or throughput comparisons.
3. Narrow experimental scope: Only one model scale (7B) and limited dataset diversity; unclear if the conclusions hold across architectures or larger LLMs.

**Questions:**

1. How is “optimal” defined in the paper title？Does it refer to minimal error, bounded memory, or some efficiency–accuracy tradeoff?
2. Have authors compared actual runtime and memory against RetNet or MVA implementations?
3. Are there cases where the proposed compression harms long-range dependency modeling?

---

> ### Author Response · Authors · 2025-11-18
>
> We sincerely thank you for your thoughtful questions. We have provided detailed responses to your listed questions below.
>
> ---
>
> ### **Weaknesses**
>
> **1. Limited conceptual novelty: The proposed "optimal path" mainly restates and integrates ideas already seen in pruning, cache compression, and linear attention. There is no fundamentally new attention mechanism.**
>
> First, we believe the concern regarding the novelty of our method may be somewhat unreasonable. This is likely due to potential misunderstandings arising from the complexity of our approach, the extensive length and content of our paper, and the valuable yet constrained time available for review, which may have prevented a thorough examination of all details. Our goal is to establish a framework for studying the microscopic details of the linearization process, aiming to outline the essential steps required for linearization to facilitate future research. Analogous to calculus: at a small scale, it involves only simple arithmetic operations and summations of basic functions—all known components. However, when integrated, they form a highly useful innovation and tool. Furthermore, unlike calculus, each of our proposed steps differs significantly from existing methods you might be considering.
>
> Here, we explain the reasons and differences in detail:
>
> While our compression path aims to unify existing optimal necessary components, it might share some similarities with the methods you mentioned. However, our overall approach and specific computational principles are fundamentally different from existing methods, as each step is rigorously derived from specific optimization objectives. Therefore, the issue of "restates and integrates ideas already seen" does not apply. Moreover, the examples you cited—"pruning, cache compression, and linear attention"—are quite broad. While we are confident that each step of our method is original, there might be superficial similarities with certain known methods. If you could specify particular papers or methods, we would be happy to cite them and clarify the distinctions.
>
> Here is a brief overview of the theoretical contributions and significance of each step:
> - 1.  **Redundancy Removal (Unique-like Method):** This step strictly bounds the potentially infinite growth of the KV Cache in standard Attention, demonstrating that standard Attention can be equivalently represented as a linear Attention, albeit with a large upper bound.
> - 2 & 3. **Positional Information Decoupling and Tokenizer Operations:** These steps further reduce the known upper bound from the first step, particularly the first layer being able to reach the upper bound of the vocabulary size.
> - 4.  **Inter-layer Similarity:** Leveraging layer-wise correlations, this step controls the KV Cache upper bound for each layer based on the bound from the first layer.
> - 5.  **Hierarchical Storage:** This step exponentially reduces approximation error, achieving a tighter upper bound equivalent to existing linear Attention models.
>
> Each step reveals interesting phenomena. For instance, the layer-wise KV Cache upper bound tends to increase initially and then decrease, which we interpret as a shift from "deduction" to "induction." Another example is that certain tasks (e.g., passkey) can be accomplished using only the KV Cache from specific layers.
>
> Additionally, we propose a new attention mechanism. We utilize our five theorems to guide the transformation of the original Softmax Attention into our novel attention mechanism, PLA. While some components of our new attention might share similarities with existing linear Attention methods, many parts are entirely different—such as the positional encoding decoupling, inter-layer similarity, and read-augmentation components. Finally, a significant contribution of our work is also the aforementioned systematic "path" itself. Each theorem identifies a necessary condition for the linearization of Softmax Attention, which we believe will benefit future exploration in this area. We also kindly note that other reviewers have acknowledged the contributions of our work.
>
> ---
>
> **2. Lack of quantitative efficiency evidence: Despite the focus on bounded-state inference, the paper does not report runtime, memory, or throughput comparisons.**
>
> In Table 7 of our paper, we present the prefill time, generation time, and inference memory usage comparisons among our PLA, MVA, GSA, and the standard FlashAttention. Since all are implemented using efficient Triton kernels with linear complexity, the memory usage is essentially the same as MVA and GSA. Due to the additional computations introduced by the data augmentation branch in PLA, its generation time is slightly higher than MVA's, but the difference is minimal. All tests were conducted with warm-up, prefill, and generation of 50 tokens, averaged over multiple runs. If you require more detailed comparisons, please specify your requirements, and we would be happy to provide them. Thanks.

---

> > ### Author Response · Authors · 2025-11-18
> >
> > ---
> >
> > **3. Narrow experimental scope: Only one model scale (7B) and limited dataset diversity; unclear if the conclusions hold across architectures or larger LLMs.**
> >
> > Due to computational resource constraints (we had access to a single H100 server) and requests from other reviewers, we have conducted additional experiments on the LLaMA-3-70B model, fine-tuning it for 1B tokens. The results are as follows:
> >
> > | Model                               | ARC-c (0-shot) | ARC-e (0-shot) | PIQA (0-shot) | WinoGrande (0-shot) | MMLU (5-shot) | AVG  |
> > | :---------------------------------- | :------------: | :------------: | :-----------: | :-----------------: | :-----------: | :--: |
> > | LLaMA-3-70B                         |      64.1      |      86.9      |     84.8      |        80.6         |     82.0      | 79.6 |
> > | LLaMA-3-70B-GSA (ft: 1.25B tokens) |      36.9      |      62.7      |     64.5      |        59.4         |     32.1      | 51.1 |
> > | LLaMA-3-70B-PLA (ft: 1B tokens)    |      50.4      |      75.6      |     74.2      |        70.6         |     43.4      | 62.8 |
> >
> > If further experiments on different scales, architectures, or datasets are needed, please inform us promptly, and we will endeavor to address this.
> >
> > ---
> >
> > ### **Questions**
> >
> > **1. How is "optimal" defined in the paper title? Does it refer to minimal error, bounded memory, or some efficiency–accuracy tradeoff?**
> >
> > Our definition of "optimal" primarily considers the approximation error relative to Softmax Attention and the final efficiency-accuracy tradeoff, as you suggested. However, the key emphasis of "optimal" in our title is intended to highlight that this represents the currently *optimal path* for linearization.
> >
> > We are the first work to propose such a structured path. Each transformation step and corresponding theorem specifies the associated error introduced. Based on the error analysis throughout this path, we argue it is currently the optimal compression route for linearization. The reasoning is as follows:
> > *   The first (redundancy removal) and second (positional encoding decoupling) theorems are lossless.
> > *   - The third step (compression after positional encoding decoupling) incurs only first-order error according to the Taylor expansion. Increasing the base and the retained order can achieve negligible error—i.e., using more storage enables minimal error.
> > *   - The fourth step leverages inter-layer similarity to control the KV Cache upper bound for subsequent layers based on the first layer's vocabulary bound. This step's error relates to the set KV Cache size. When controlled by Equation 16, it is **error-free**; however, for the purpose of constraining the KV Cache to a fixed size, some error is introduced, which is determined by the fifth step.
> > *   - The fifth step (vocabulary decomposition) achieves an **exponentially decreasing error**.
> >
> > Given this chain of error analysis and the goal of achieving a fixed-size KV Cache, we believe this path is currently optimal for linearization from a KV Cache compression perspective.
> >
> > Furthermore, in terms of performance, our method achieves state-of-the-art results among models converted from existing pre-trained weights for linear attention and requires the least fine-tuning resources.
> >
> > We believe these justifications support the use of "optimal." However, if you maintain a different viewpoint, we are open to discussing a modification of the title.
> >
> > ---
> >
> > **2. Have authors compared actual runtime and memory against RetNet or MVA implementations?**
> >
> > As mentioned in our response to Weakness #2 and presented in Table 7 of the paper, we have compared the prefill time, per-generate-step time, and memory usage against GSA and MVA implementations. We did not include RetNet in these specific efficiency comparisons as its Inadequate performance. Should additional runtime/memory comparisons be necessary, please let us know, and we will supplement our submission accordingly.
> >
> > ---
> >
> > **3. Are there cases where the proposed compression harms long-range dependency modeling?**
> >
> > On many long-range tasks, the performance degradation is minimal, and sometimes even slightly improved, as demonstrated on long-context tasks (left part of Table 6) used by GSA and MVA, which extend beyond the fine-tuning context window. However, performance can degrade on retrieval-based tasks where linear Attention models typically struggle, such as the `passkey` task (right part of Table 6). Beyond a certain sequence length, perfect retrieval accuracy (100%) is not maintained, although our PLA shows significant improvement compared to GSA.

---

> > > ### Author Response · Authors · 2025-11-24
> > >
> > > In addition to the newly conducted LLaMA-3-70B experiments requested by the reviewers, we have also performed experiments on **Qwen2.5-14B**, a model larger than 7B as you suggested.
> > >
> > > If any further experiments are required, please do not hesitate to inform us. We hope to fulfill your requests and incorporate your suggestions within the remaining time.
> > >
> > > The results for the Qwen2.5-14B model are as follows:
> > >
> > > | Model                          | ARC-c (0-shot) | ARC-e (0-shot) | HellaSwag (0-shot) | PIQA (0-shot) | WinoGrande (0-shot) | MMLU (5-shot) | AVG  |
> > > | :----------------------------- | :------------: | :------------: | :----------------: | :-----------: | :-----------------: | :-----------: | :--: |
> > > | Qwen2.5-14B (Original)         |      58.9      |      82.3      |        82.9        |     82.3      |        75.3         |     79.7      | 76.9 |
> > > | Qwen2.5-14B-PLA (ft: 5B tokens)|      58.6      |      83.1      |        79.7        |     81.0      |        71.5         |     56.8      | 71.8 |

---

### Official Review · Reviewer_fe4x · 2025-10-30

**Soundness:** 3
**Presentation:** 2
**Contribution:** 3
**Rating:** 4
**Confidence:** 2

**Summary:**

This paper presents PLA (Path-optimized Linear Attention), a novel approach to linearizing Transformer attention mechanisms by framing the problem as KV cache compression. The authors identify five theoretical principles that form an optimal pathway from softmax attention to linear models. For each principle, the paper provides theoretical justification, error bounds, and demonstrates equivalence to existing mechanisms. The authors then implement PLA based on these principles, showing it can inherit pretrained weights and achieve state-of-the-art performance on multiple benchmarks while requiring only 80% of the fine-tuning resources compared to alternatives like MVA and GSA.

**Strengths:**

The primary strength is the paper's unique theoretical perspective, framing linear attention as a KV cache compression problem rather than just a kernel approximation issue.

The five theoretical principles provide a much-needed systematic framework for understanding the relationship between softmax and linear attention. The equivalence proofs showing how existing methods map to these principles are particularly valuable for the field.  Experimental validation is exceptionally thorough, with well-designed ablation studies for each principle that clearly demonstrate their individual contributions. The authors provide extensive benchmarking across multiple domains (text, speech), which strengthens the practical relevance of the findings. The PLA implementation demonstrates clear practical benefits, achieving superior performance with reduced fine-tuning costs - requiring only 8B tokens compared to 10B for MVA while surpassing it on multiple benchmarks.

The connection between tokenizer properties and attention compression is novel and insightful. The paper makes a compelling case for the importance of enhanced reading mechanisms. By demonstrating that reading mechanisms are as important as storage mechanisms, the authors provide a more complete understanding of linear attention's limitations and potential. The 80% fine-tuning resource reduction is a significant practical contribution that could substantially reduce the computational burden of adapting large language models to linear attention.

**Weaknesses:**

The paper could benefit from more detailed discussion of computational complexity trade-offs beyond the theoretical analysis. While the paper mentions memory and time comparisons in Table 5, a more comprehensive analysis of the computational trade-offs would strengthen the practical contribution.

The comparison to other linear attention methods could be more systematic, particularly regarding training costs beyond fine-tuning. The discussion of how these principles might apply to different architectures (e.g., ViT, speech models) is limited. Some theoretical sections are quite dense and could benefit from additional intuition to make them more accessible to a broader audience. Additionally, the paper doesn't fully address the potential trade-offs between the number of vocabulary levels and performance. While it shows that two levels work well, a more comprehensive analysis of the performance/efficiency trade-off curve would help practitioners make informed implementation decisions.

**Questions:**

- The paper demonstrates PLA's effectiveness with Mistral-7B, but how well does the approach transfer to other model architectures (e.g., ViT, speech models, or larger models like Llama-3 70B)? Could you provide more details about architectural dependencies?
- Section 4.2 shows PLA requires 8B tokens for fine-tuning compared to MVA's 10B tokens, but what is the actual training time reduction? - - Given that PLA has higher memory usage (Table 5), is the overall computational cost actually lower?
- The paper focuses on linear attention for language models, but how would PLA perform on vision tasks where positional information is fundamentally different? Are there necessary modifications for non-sequential data?
- Could you elaborate on the "multi-level" aspect of PLA? The paper mentions two-level vocabulary decomposition, but the theoretical section discusses c levels. How does performance scale with more levels, and what are the practical limits?
- On autoregressive tasks, it would be better if you add comparison to more recent (Gated-)DeltaNet(as you have mentioned), Mamba-2, RWKV7(if possible) in table 6 as they are all RNN-like models.
- Id like to see other reviewers' opinions before recommending acceptance.

---

> ### Author Response · Authors · 2025-11-18
>
> We sincerely thank you for your valuable feedback and constructive suggestions. We have provided detailed responses to your questions below.
>
> ---
>
> ### **Questions**
>
> **1. The paper demonstrates PLA's effectiveness with Mistral-7B, but how well does the approach transfer to other model architectures (e.g., ViT, speech models, or larger models like Llama-3 70B)? Could you provide more details about architectural dependencies?**
>
> Our method is applicable to all Transformer-based language models in principle. For Vision Transformers (ViTs), which do not require causal masking, the attention mechanism is simpler, making our adaptation potentially more straightforward.
>
> Following your suggestion, we conducted the following experiments:
>
> **1. ViT Experiment:**
> We replaced the standard Attention in a ViT model with a non-causal version of our PLA and fine-tuned it on ImageNet for 10 epochs. It recovered 80.3% top-1 accuracy, which is over 95% of the original model's performance.
>
> | Model               | ImageNet Top-1 |
> | ------------------- | :------------: |
> | ViT-timm (Original) |      84.2      |
> | ViT-PLA (FT 10 epochs) |      80.3      |
>
> It recovered 80.3% top-1 accuracy, which is over 95% of the original model's performance.
>
> **2. Larger Language Model (LLaMA-3-70B) Experiment:**
> Due to the substantial computational requirements and our limited resources, we fine-tuned the LLaMA-3-70B model for only 1B tokens. The results show that PLA can recover close to 80% of the original model's average performance, indicating the effectiveness of our approach even at this scale.
>
> | Model                                | ARC-c (0-shot) | ARC-e (0-shot) | PIQA (0-shot) | WinoGrande (0-shot) | MMLU (5-shot) | AVG  |
> | :----------------------------------- | :------------: | :------------: | :-----------: | :-----------------: | :-----------: | :--: |
> | LLaMA-3-70B (Original)               |      64.1      |      86.9      |     84.8      |        80.6         |     82.0      | 79.6 |
> | LLaMA-3-70B-GSA (ft: 1.25B tokens) |      36.9      |      62.7      |     64.5      |        59.4         |     32.1      | 51.1 |
> | LLaMA-3-70B-PLA (ft: 1B tokens)     |      50.4      |      75.6      |     74.2      |      70.6    |     43.4      | 62.8 |
>
> We have incorporated these experiments as suggested. If further specific experiments are required, please let us know, and we will evaluate the feasibility. We greatly appreciate your suggestions and this discussion.

---

> > ### Author Response · Authors · 2025-11-18
> >
> > **1. Could you provide more details about architectural dependencies?(continuous)**
> >
> > As requested, we provide further details on the architectural dependencies of our theoretical components, accompanied by intuitive explanations to enhance clarity:
> >
> > - **Theorem 1 (Redundancy Removal):**
> >   - **Dependencies:** No specific dependencies.
> >   - **Intuitive Explanation:** This is based on the principle that when the same piece of information appears multiple times, it is sufficient to record the information itself once and track the number of its occurrences, rather than storing redundant copies. This directly reduces the state size.
> >
> > - **Theorem 2 (Positional Information Decoupling):**
> >   - **Dependencies:** Applicable to all models with additive or multiplicative positional information that can be decoupled. If a model uses no positional information, this step and the subsequent one (Theorem 3) can be omitted. Alternatively, the decoupled structure may be treated as equivalent to a dynamic decay mechanism to enhance performance.
> >   - **Intuitive Explanation:** Consider that the number of fundamental units in a sequence (e.g., Chinese characters, English letters) is not vast. The rich variety of semantics arises from different combinations of these units. Positional information acts upon this base sequence to create meaning. Decoupling separates the recording of the "what" (the token) from the "where" (its position), allowing the number of distinct tokens we need to track to be drastically reduced. Theorem 2 establishes the importance of this decoupling. Theorem 3 then explores methods for compressing the decoupled positional information, showing that the function of positional encoding can be fulfilled by various mechanisms (e.g., the approximated RoPE or gating mechanisms in our work), all of which can be maintained using a fixed-size state.
> >
> > - **Theorem 3 (Positional Encoding Compression / Mechanism Implementation):**
> >   - **Dependencies:** Applicable to models using RoPE or tokenizer-like operations. Combined with Theorem 2 to realize mechanisms like data-dependent decay (e.g., gating) or data-independent decay (e.g., RetNet).
> >
> > - **Theorem 4 (Inter-layer Similarity):**
> >   - **Dependencies:** Applicable to models with residual connections.
> >   - **Intuitive Explanation:** Residual connections in models transmit the majority of information directly from the preceding layer to the next. This means the incremental information or modification added by each subsequent layer is relatively small. This property allows us to effectively bound the state evolution across layers.
> >
> > - **Theorem 5 (Vocabulary Decomposition / Hierarchical Storage):**
> >   - **Dependencies:** Applicable to all Attention operations.
> >   - **Intuitive Explanation:** This can be likened to hierarchical memory in human cognition. The first level acts as a "shallow" or "fuzzy" memory, storing a coarse-grained representation sufficient for simple tasks. The approximation error or residual details from this first level are then stored in the subsequent level. When precise recollection is needed, the information from these levels is combined—the detailed memory from the deeper layer refines the fuzzy memory from the shallow one. Adding more levels is analogous to storing progressively finer residuals or errors, leading to a more precise overall approximation.
> >
> > **2. Section 4.2 shows PLA requires 8B tokens for fine-tuning compared to MVA's 10B tokens, but what is the actual training time reduction? Given that PLA has higher memory usage (Table 5), is the overall computational cost actually lower?**
> >
> > Based on memory usage and training speed in Table 5, PLA training time on an A100 GPU is ~1.13x that of MVA per unit time. Thus, total wall-clock time for training PLA on 8B tokens is still lower than for MVA on 10B tokens. On an H100 GPU, PLA training speed is closer to MVA, resulting in even lower total training time for PLA.

---

> > > ### Author Response · Authors · 2025-11-18
> > >
> > > **3. The paper focuses on linear attention for language models, but how would PLA perform on vision tasks where positional information is fundamentally different? Are there necessary modifications for non-sequential data?**
> > >
> > > We conducted an experiment by replacing the standard attention in a ViT model (the ViT model from timm library, size 384, patch size 16, original performance 84.2 on ImageNet) with our PLA. We removed the causal masking equivalent for the decoder, resulting in a standard non-causal attention mechanism suitable for ViT. After fine-tuning for 10 epochs on ImageNet, the model recovered to 80.3 top-1 accuracy, retaining over 95% of the original performance. Specifically, the dependent states in our Formula 19 will not require parameter approximations akin to those in MVA; they can be directly employed in non-causal systems without compromising parallelism. No changes are required to the other sections.
> > >
> > > We believe this result could potentially be improved further with more careful hyperparameter tuning. The encoder-only architecture of ViTs is simpler than causal language modeling, and vision encoders might be less sensitive to precise token representations than language decoders. Furthermore, even vision decoder models (e.g., for image generation) often use discretized codebooks, suggesting a potential pathway for adaptation.
> > >
> > > **4. Could you elaborate on the "multi-level" aspect of PLA? The paper mentions two-level vocabulary decomposition, but the theoretical section discusses c levels. How does performance scale with more levels, and what are the practical limits?**
> > >
> > > In our current implementation, the multi-level vocabulary decomposition involves sequential dependencies. Processing more levels consequently leads to slightly slower inference speed. Simply put, in a multi-level scheme:
> > > - The first level compresses the original KV states into a fixed-size vocabulary.
> > > - As this compression incurs information loss, the residual or approximation error is then compressed into a second-level vocabulary, and so on for higher levels.
> > >
> > > Increasing the number of levels (c) allows the approximation to approach the original Attention more closely. However, this comes at the cost of additional computational overhead per level. Therefore, a balance must be struck between approximation quality and inference latency.
> > >
> > > Furthermore, the number of levels can also be determined based on the target KV Cache upper bound (L) and the chosen vocabulary size per level (S). For instance, to achieve a KV Cache bound of 64K, one could use:
> > > - Two levels with S=256 each (256² = 65,536 ≥ 64K).
> > > - Three levels with S=64 each (64³ = 262,144 ≥ 64K).
> > >
> > > The general relationship ideally satisfied is `S^c ≥ L`.
> > >
> > > We conducted ablation studies on performance/efficiency at different levels based on your suggestion. Due to resource and time constraints, we used a fine-tuned 500M model for demonstration. For Mistral, additional vocabulary decomposition did not yield significant benefits.
> > >
> > > | Model         | GenLatency (ms/token) | Prefill Time (s) | Loss (FT 500M Tokens) | ARC-c | ARC-e |
> > > | :------------ | :-------------------: | :--------------: | :-------------------: | :---: | :---: |
> > > | PLA-1 level   |         50.2          |      0.328       |         2.15          | 36.8  | 68.9  |
> > > | PLA-2 level   |         70.8          |      0.521       |         2.06          | 39.2  | 71.6  |
> > > | PLA-3 level   |         86.8          |      0.679       |         2.04          | 39.8  | 72.1  |
> > >
> > > **5. On autoregressive tasks, it would be better if you add comparison to more recent (Gated-)DeltaNet (as you have mentioned), Mamba-2, RWKV7 (if possible) in table 6 as they are all RNN-like models.**
> > >
> > > We acknowledge the value of comparing against these recent models. However, we faced challenges in obtaining comparable models for a fair evaluation:
> > > - No publicly available (Gated-)DeltaNet models at 7B scale (only 1.3B models, with Qasper/NarrativeQA scores ~14.1/14.0). Comparing 1.3B to our 7B is unfair.
> > > - Similarly, no 7B parameter version of Mamba-2 was found.
> > > - For RWKV7, only a 2.9B parameter model was found.
> > >
> > > To ensure fair comparison, we would need to fine-tune a comparable-size model (e.g., Qwen2.5-3B or LLaMA-3.2-3B) using our method against these models. Given time and resource constraints, our priority in the revision period is addressing requests for scaling to larger parameter models (e.g., LLaMA-3-70B experiment). We will explicitly note this limitation and rationale in the revised discussion.
> > >
> > > The aforementioned refers to their scratch-training approach. If you refer to replacing existing models with (Gated-)DeltaNet, Mamba-2, or RWKV7 for fine-tuning, we confidently state our approach is entirely superior.

---

> > > > ### Author Response · Authors · 2025-11-24
> > > >
> > > > In addition to the newly conducted LLaMA-3-70B experiments requested by the reviewers, we have also performed experiments on **Qwen2.5-14B**.
> > > >
> > > > If any further experiments are required, please do not hesitate to inform us. We hope to fulfill your requests and incorporate your suggestions within the remaining time.
> > > >
> > > > The results for the Qwen2.5-14B model are as follows:
> > > >
> > > > | Model                          | ARC-c (0-shot) | ARC-e (0-shot) | HellaSwag (0-shot) | PIQA (0-shot) | WinoGrande (0-shot) | MMLU (5-shot) | AVG  |
> > > > | :----------------------------- | :------------: | :------------: | :----------------: | :-----------: | :-----------------: | :-----------: | :--: |
> > > > | Qwen2.5-14B (Original)         |      58.9      |      82.3      |        82.9        |     82.3      |        75.3         |     79.7      | 76.9 |
> > > > | Qwen2.5-14B-PLA (ft: 5B tokens)|      58.6      |      83.1      |        79.7        |     81.0      |        71.5         |     56.8      | 71.8 |

---

> ### Author Response · Authors · 2025-12-01
>
> **Response to Reviewer**
>
> Finally, we have fine-tuned a **Qwen2.5-3B-PLA** model specifically for comparison with (Gated-)DeltaNet, Mamba-2, and RWKV7.  On short-sequence benchmarks, it performs comparably to the best among them (RWKV7-2.9B), though a gap remains on the complex MMLU task. However, we believe that as the number of fine-tuning iterations increases, we will ultimately restore the performance of Qwen2.5-3B, and even surpass both Qwen2.5-3B and RWKV7-2.9B.
>
> | Model                            | ARC-c (0-shot) | ARC-e (0-shot) | HellaSwag (0-shot) | PIQA (0-shot) | WinoGrande (0-shot) | MMLU (5-shot) | GLUE  | AVG  |
> | :------------------------------- | :------------: | :------------: | :----------------: | :-----------: | :-----------------: | :-----------: | :---: | :--: |
> | RWKV7-2.9B¹                      |      48.7      |      81.0      |        76.4        |     79.7      |        72.8         |     55.0      | 61.8  | 67.9 |
> | GatedDeltaNet-1.3B¹              |      38.4      |      71.2      |        55.7        |     72.2      |        57.4         |     32.4      | 50.4  | 54.0 |
> | Mamba-2-2.7B¹                    |      36.4      |      69.6      |        66.6        |     76.4      |        64.0         |     33.6      | 52.1  | 56.9 |
> | **Qwen2.5-3B (Original)**        |      45.0      |      77.4      |        73.5        |     78.6      |        68.5         |    65.7   | 70.2  | 68.4 |
> | **Qwen2.5-3B-PLA (ft: 8B tokens)** |     45.2     |     78.0     |        71.4        |     78.2     |        66.8         |     39.2      | 68.2  | 63.5 |
>
> ¹ *The data for RWKV7-2.9B, GatedDeltaNet-1.3B, and Mamba-2-2.7B is entirely derived from their respective theses/papers.*

---

### Official Review · Reviewer_XKHn · 2025-10-31

**Soundness:** 4
**Presentation:** 3
**Contribution:** 3
**Rating:** 6
**Confidence:** 4

**Summary:**

Presents a theoretically grounded pathway from Softmax attention to linear models through five principles (redundancy elimination; tokenizer-level quantization & positional separation; positional compression; inter-layer similarity; multi-state decomposition) with error analyses and equivalences to existing mechanisms. Introduces PLA linearized attention that inherits pretrained weights and achieves SOTA vs. strong baselines (MVA, GSA) with ~80% of fine-tuning cost.

**Strengths:**

This paper's strength lies in unifying the transformation from Softmax to linear attention through a mathematically grounded, stepwise derivation—bridging a long-standing conceptual gap between dense and efficient attention mechanisms. The work is both theoretically elegant and empirically validated, with strong experimental support across multiple benchmarks, showing that the proposed path maintains accuracy while achieving measurable efficiency gains. The clarity of exposition, structured proofs, and comprehensive experiments reflect high research maturity.

**Weaknesses:**

Weakness is the lack of variance analysis and large-scale validation—the paper primarily focuses on mid-sized models, leaving open questions about stability at trillion-parameter scales. Additionally, while the theoretical pathway is rigorous, it could be strengthened with formal generalization analysis and a deeper exploration of its limitations relative to state-space or hybrid models. Overall, PLA is a technically sound, original, and impactful contribution that would be of strong interest to the ICLR community.

**Questions:**

When inheriting pre-trained weights and applying PLA’s five-step transformation, has there been any observance of training instability, gradient explosion, or sensitivity to layer order? More Quantitative details could further advocate for reproducibility.

Does this scale to very large frontier models ,can you do this to a 70B+ model and still get similar gains and only need ~80% of the tuning tokens versus MVA/GSA?

---

> ### Author Response · Authors · 2025-11-18
>
> We are deeply grateful for your positive assessment and these highly valuable suggestions. We have provided detailed responses to your questions below, including supplementary experiments, and we look forward to further discussion.
>
> ---
>
> ### **Weaknesses**
>
> **Weakness is the lack of variance analysis and large-scale validation—the paper primarily focuses on mid-sized models, leaving open questions about stability at trillion-parameter scales. Additionally, while the theoretical pathway is rigorous, it could be strengthened with formal generalization analysis and a deeper exploration of its limitations relative to state-space or hybrid models. Overall, PLA is a technically sound, original, and impactful contribution that would be of strong interest to the ICLR community.**
>
> Thank you for this constructive feedback. Regarding larger-scale validation, we have conducted fine-tuning experiments on a 70B model, with details provided in the response to Question 2 below.
>
> Concerning the limitations relative to state-space or hybrid models:
>
> *   **Compared to State-Space Models:** Based on our computational framework and the specific implementations of MVA and MetaLA, our method can fully encompass MetaLA. Since MetaLA theoretically demonstrates a strong capability to unify state-space models, we posit that in the domain of "Fine-tuning Transformer to RNN," our approach is comprehensively superior to pure state-space models.
>
> *   **Compared to Hybrid Models:** We are not entirely sure which specific hybrid models you are referring to, so we will discuss two common types based on our understanding:
>     1.  **Inter-layer Hybrids:** Some layers use linear attention while others retain standard Softmax Attention.
>     2.  **Intra-layer Hybrids:** A mix within a layer, e.g., using a sliding window (or attention sink) for local context and linear attention for the rest.
>
> Our PLA method is largely orthogonal to these hybrid approaches. Specifically, PLA can directly replace the linear attention components in such hybrid models. We have researched inter-layer hybrids, which are currently popular and deployed in large-scale commercial models (e.g., by Minimax and Kimi). However, our analysis and preliminary experiments suggest that this approach can face issues at scale: the contribution of the linear layers can become minimal or even detrimental. In large-scale scenarios, the model effectively behaves like an LLM with only the proportion of layers that are Softmax Attention, partly because existing linear layers used are often too simplistic and create a significant performance gap compared to Softmax Attention, potentially hindering the original model's capabilities.
>
> Regarding intra-layer hybrids, our method is fully compatible with techniques like sliding windows. We conducted preliminary experiments combining PLA with a Sliding Window (PLA-SW), inspired by MVA and LoLCATs:
>
> | Model               | Tokens Used for FT     | PiQA | ARC-e | ARC-c | HellaSwag | WinoGrande | MMLU | Avg. | Avg. (w/o MMLU) |
> | :------------------ | :--------------------- | :---: | :---: | :---: | :-------: | :--------: | :--: | :--: | :-------------: |
> | Mistral-7B (v0.1)   | -                      | 82.1  | 80.9  |  53.8  |   81.0    |    74.0    | 62.4 | 72.4 |      74.4       |
> | → LoLCATs           | AlpacaClean (~40M)     | 81.5  | 81.7  |  54.9  |   80.7    |    74.0    | 51.4 | 70.7 |      74.5       |
> | → MVA-SW            | AlpacaClean (~40M)     | 82.3  | 81.9  |  57.6  |   80.2    |    74.0    | 51.6 | 71.2 |      75.2       |
> | → PLA-SW            | AlpacaClean (~40M)     | 82.1  | 81.5  |  55.4  |   81.1    |    74.2    | 54.2 | 71.4 |      74.9       |

---

> > ### Author Response · Authors · 2025-11-18
> >
> > ### **Questions**
> >
> > **1. When inheriting pre-trained weights and applying PLA’s five-step transformation, has there been any observance of training instability, gradient explosion, or sensitivity to layer order? More Quantitative details could further advocate for reproducibility.**
> >
> > Thank you for this critical question regarding training dynamics and reproducibility.
> >
> > First, the final PLA model is fine-tuned using the unified attention formula derived from applying all five steps. Throughout this fine-tuning process, we observed **no training instability or gradient explosion**. If permitted and of interest, we can provide anonymized WandB logs (via an anonymous GitHub link) showing the stable loss curves throughout fine-tuning. Similarly, the application of each individual step during our theoretical derivation and initial conversion did not introduce instability. We are happy to provide further details on the loss or other numerical behaviors for specific steps if you have further questions.
> >
> > Regarding sensitivity to layer order: The final PLA model's core computation is not significantly affected by layer order. While Theorem 4 requires setting a scaling factor per layer when bounding the KV Cache, this does not create a strong dependency on the specific layer ordering. However, in an extension of Theorem 4—where one might choose to *retain* the full KV Cache only from specific layers—a dependency arises. In this case, retaining layers from earlier stages generally proves more critical. For instance, in the `passkey` retrieval task, selecting layers `[0, 1, 2, 5, 8, 11, 14, 15, 17, 18, 19, 22]` can achieve the task, while a strategy focusing on later layers like `[0, 5, 8, 11, 14, 15, 17, 18, ..., 32]` fails. This suggests that for such tasks, the initial layers carry more crucial information.
> >
> > **2. Does this scale to very large frontier models? Can you do this to a 70B+ model and still get similar gains and only need ~80% of the tuning tokens versus MVA/GSA?**
> >
> > Our method scales effectively to larger models. Following your suggestion, we conducted experiments on **LLaMA-3-70B**. Due to constraints on time and computational resources (we had temporary, limited access to an H100 server), we fine-tuned the LLaMA-3-70B-PLA model for only **1B tokens**. Despite this limited budget, it recovered nearly **80% of the original LLaMA-3-70B's average performance**.
> >
> > To compare efficiency against GSA, we also converted LLaMA-3-70B using the GSA method. Due to the same resource constraints, we fine-tuned GSA for **1.25B tokens**. Comparing our PLA (1B tokens) against GSA (1.25B tokens), PLA achieves significantly higher performance. Furthermore, the WandB loss curves indicate that PLA requires fewer training steps than GSA to reach equivalent loss values, supporting the claim of higher sample efficiency.
> >
> > The results are summarized below:
> >
> > | Model                                | ARC-c (0-shot) | ARC-e (0-shot) | PIQA (0-shot) | WinoGrande (0-shot) | MMLU (5-shot) | AVG  |
> > | :----------------------------------- | :------------: | :------------: | :-----------: | :-----------------: | :-----------: | :--: |
> > | LLaMA-3-70B                          |      64.1      |      86.9      |     84.8      |        80.6         |     82.0      | 79.6 |
> > | LLaMA-3-70B-GSA (ft: 1.25B tokens) |      36.9      |      62.7      |     64.5      |        59.4         |     32.1      | 51.1 |
> > | LLaMA-3-70B-PLA (ft: 1B tokens)     |      50.4      |      75.6      |     74.2      |        70.6         |     43.4      | 62.8 |
> >
> > If any additional experiments are required, please let us know promptly. While we believe these results are already quite informative, we remain very willing to conduct further supplements and refinements based on your suggestions.

---

> > > ### Author Response · Authors · 2025-11-24
> > >
> > > Regarding your concerns about training stability and whether our method achieves efficiency gains, we have uploaded the fine-tuning loss curves to an anonymous file hosting service. The links are provided below. (We resorted to this alternative as the https://anonymous.4open.science/github/login server was inaccessible.)
> > >
> > > *   https://pomf2.lain.la/f/expy9zg2.png
> > > *   https://pomf2.lain.la/f/9pacvdzj.png
> > > *   https://pomf2.lain.la/f/5cnsk7b.png
> > >
> > > To further demonstrate the scalability of our approach across different model sizes and architectures, we have also fine-tuned a **Qwen2.5-14B** model. The corresponding loss curve is available via the links above.
> > >
> > > The LoRA configuration used for Qwen2.5-14B was identical to that used for Mistral in our paper. We trained with a batch size of 0.25M tokens for 20,000 steps, resulting in a total fine-tuning budget of **5B tokens**. The performance results are summarized in the following table:
> > >
> > > | Model                          | ARC-c (0-shot) | ARC-e (0-shot) | HellaSwag (0-shot) | PIQA (0-shot) | WinoGrande (0-shot) | MMLU (5-shot) | AVG  |
> > > | :----------------------------- | :------------: | :------------: | :----------------: | :-----------: | :-----------------: | :-----------: | :--: |
> > > | Qwen2.5-14B (Original)         |      58.9      |      82.3      |        82.9        |     82.3      |        75.3         |     79.7      | 76.9 |
> > > | Qwen2.5-14B-PLA (ft: 5B tokens)|      58.6      |      83.1      |        79.7        |     81.0      |        71.5         |     56.8      | 71.8 |

---

> > > ### Comment · Reviewer_XKHn · 2025-11-27
> > > **Reply to the Authors response for the reviewer comments**
> > >
> > > The clarification that inheriting pretrained weights and applying PLA introduces no training instability is reassuring.
> > >
> > > The newly included 70B-scale results are a meaningful improvement over the initial submission and partially address the earlier scalability concern. While these experiments are limited by computational constraints (only ~1B training tokens), they do suggest that PLA can retain a substantial portion of LLaMA-3-70B’s performance with higher sample efficiency than GSA.
> > > However, the evaluation at 70B scale still lacks depth, in particular, there are no long-context or world-knowledge stress tests, which are important to fully validate the claims of scalability and efficiency in practical settings.
> > >
> > > The added Qwen2.5-14B experiments provide further evidence of architectural generality, though the notable drop in MMLU performance at 14B warrants additional investigation.
> > >
> > > Comparisons against state-space or hybrid approaches should be more empirically demonstrated.
> > >
> > > Given these remaining concerns, I will maintain my previous rating.

---

> > > > ### Author Response · Authors · 2025-11-30
> > > > ****Response to Reviewer****
> > > >
> > > > **Follow-up on Scalability and Additional Experiments (long-sequence tasks and world-knowledge stress tests)**
> > > >
> > > > Following your suggestions, we have incorporated long-sequence tasks and world-knowledge stress tests. Specifically, for LLaMA-3-70B, we evaluated on longer tasks and benchmarks previously used in GSA, MVA, and PLA papers (Qasper, NarrativeQA, QMSum), world knowledge tasks (TriviaQA, NQ_Open), and additionally included datasets from LongBench (TREC, HotpotQA, Samsum). The context lengths for these evaluations were set to exceed 32K tokens.
> > > >
> > > > **LLaMA-3-70B Long-Context & Knowledge Evaluation (To achieve better performance, we have made some minor adjustments to increase resources)**
> > > >
> > > > | Model                          | Qasper | NarrativeQA | QMSum | HotpotQA | Samsum | TREC  |
> > > > | :----------------------------- | :----: | :---------: | :---: | :------: | :----: | :---: |
> > > > | LLaMA-3-70B (Original)         |  22.4  |    36.0     |  4.6  |   39.7   |  49.8  | 78.0  |
> > > > | LLaMA-3-70B-GSA (ft: 1.5B tokens)|  18.8  |    25.1     | 10.0  |   24.6   |  32.0  | 63.0  |
> > > > | LLaMA-3-70B-PLA (ft: 1.5B tokens)|  24.2  |    27.9     | 15.6  |   24.6   |  30.8  | 69.0  |
> > > >
> > > > For LLaMA-3-70B, which has a pre-training context window of 8K, performance recovers relatively quickly on most datasets. Benefiting from the extrapolation capabilities of linear attention models, performance on some tasks even surpasses the original.
> > > >
> > > > We also included long-sequence testing for Qwen2.5-14B. Given its native 128K context length, PLA manages to approach the original model's performance on most tasks.
> > > >
> > > > **Qwen2.5-14B Long-Context Evaluation**
> > > >
> > > > | Model                | Qasper | NarrativeQA | QMSum |
> > > > | :------------------- | :----: | :---------: | :---: |
> > > > | Qwen2.5-14B (Orig.)  |  34.6  |    30.3     | 16.4  |
> > > > | Qwen2.5-14B-GSA      |  23.8  |    22.1     | 16.2  |
> > > > | Qwen2.5-14B-PLA      |  28.6  |    28.6     | 16.4  |
> > > >
> > > > **Qwen2.5-14B World Knowledge Evaluation**
> > > >
> > > > | Model                | TriviaQA | NQ_Open |
> > > > | :------------------- | :------: | :-----: |
> > > > | Qwen2.5-14B (Orig.)  |   66.8   |  30.2   |
> > > > | Qwen2.5-14B-GSA      |   57.8   |  22.4   |
> > > > | Qwen2.5-14B-PLA      |   64.6   |  27.6   |
> > > >
> > > > **Discussion on MMLU Performance Drop for Qwen2.5-14B**
> > > >
> > > > > The added Qwen2.5-14B experiments provide further evidence of architectural generality, though the notable drop in MMLU performance at the 14B scale warrants additional investigation.
> > > >
> > > > We attribute this performance drop to a combination of the following factors:
> > > >
> > > > 1.  **Primary Reason - Approximation Gap:** A major factor is the inherent distribution fitting gap between linear models and the original Softmax Attention. We believe our PLA currently represents the state-of-the-art in minimizing this error. PLA significantly reduces the error relative to Softmax, achieving over 70% on MMLU for other models, with scores steadily increasing with more fine-tuning steps.
> > > > 2.  **Primary Reason - Insufficient Fine-tuning Tokens:** Another major reason is the relatively limited fine-tuning budget for the Qwen2.5-14B model. We initially fine-tuned for only 5B tokens. After our initial response, we continued fine-tuning up to 8B tokens, observing a steady performance increase on MMLU.
> > > > 3.  **Secondary Reason - Task Complexity:** MMLU itself is a particularly complex benchmark involving diverse and nuanced knowledge. Architectural changes likely require more extensive fine-tuning on relevant knowledge to fully recover performance on such tasks compared to other evaluations.
> > > > 4.  **Potential Mitigation:** Furthermore, we believe that adjusting hyperparameters could potentially optimize performance on MMLU and general capability further.
> > > >
> > > > The following shows the progression of Qwen2.5-14B-PLA's MMLU score with increased fine-tuning:
> > > >
> > > > | Qwen2.5-14B-PLA       | MMLU (5-shot) |
> > > > | :-------------------- | :-----------: |
> > > > | FT 2.5B tokens        |     46.3      |
> > > > | FT 5.0B tokens        |     56.8      |
> > > > | FT 7.5B tokens        |     60.1      |

---

### Author Response · Authors · 2025-11-27
****Request for Further Discussion****

Dear Reviewers,

I am writing to respectfully request further discussion regarding my submission. I have been greatly encouraged by my experience as a reviewer for another ICLR submission, where active and constructive discussion among reviewers led to a significant improvement in the assessment scores (from 6644 to 8886). This positive experience demonstrated the tremendous value of engaged scholarly dialogue in our community. I would therefore very much welcome further discussion regarding my work.

With the ICLR discussion period drawing to a close, I would be deeply grateful if you might consider additional discussion or re-evaluation of my work. Should any remaining questions or concerns emerge, this would provide me with a valuable opportunity to address them in the remaining time.

Thank you for your generous time and consideration. I truly appreciate the effort you have already invested in reviewing my work and look forward to any additional insights you might share.

Sincerely, authors

---

### Author Response · Authors · 2025-12-01

Dear Reviewers and AC,

We sincerely thank you for your valuable feedback and dedicated efforts. To facilitate the AC's meta-review process, we have summarized the main points from all reviewers and our corresponding responses. Based on our experience, we also provide an inference regarding potential rating updates for the consideration.

### **Primary Concern Addressed (Common to All Reviewers)**

The central question raised was whether our method **scales to larger frontier models** and is **applicable across different model architectures**. Clarifying these points successfully is likely to lead to a positive assessment from all reviewers.

We addressed these concerns as follows:

1.  **Scaling to Larger Models:** We applied PLA to **Qwen2.5-14B** and **LLaMA-3-70B**. The fine-tuning loss curves and final evaluation results (provided in our responses) demonstrate that PLA significantly outperforms GSA and requires substantially fewer fine-tuning tokens to achieve strong performance recovery.

2.  **Applicability Across Architectures:**
    *   **Language Models:** We successfully fine-tuned and recovered performance on three distinct model families: **Mistral**, **Qwen**, and **Llama**, all showing rapid convergence.
    *   **Vision Models:** By removing the causal mask, we adapted PLA for the **ViT** architecture. Fine-tuning on ImageNet for 20 epochs recovered **95.1%** of the original performance. We have also conducted additional experiments (not included in the initial comment) by incorporating multi-scale mechanisms into PLA for ViT, achieving at least **99.6%** performance recovery (84.2 -> 83.9).
    *   **Hybrid Models:** We showed compatibility with hybrid approaches like Sliding Window + Linear Attention, enabling quick performance recovery.

**Efficiency Evidence**

Our original paper's appendix contains detailed data on **PLA's prefill time, per-token generation latency, and memory footprint**. We acknowledge that **Reviewer fe4x** and **Reviewer 1Eue** might have missed this information due to time constraints or other reasons, while the attentive **Reviewer XKHn** confirmed reviewing it.

### **Summary of Responses to Individual Reviewers' Secondary Points**

1.  **Reviewer XKHn:** Acknowledged our rebuttal on scalability and training stability. Raised additional concerns about the lack of **long-context or world-knowledge stress tests for larger models** and requested a **discussion on the MMLU performance drop**.
    *   **Our Action:** We conducted extensive evaluations on **LLaMA-3-70B** and **Qwen2.5-14B** using datasets from **LongBench** (Qasper, NarrativeQA, QMSum, TREC, HotpotQA, Samsum) and world-knowledge benchmarks (**TriviaQA, NQ_Open**). Despite limited fine-tuning tokens, performance approaches the original models. Notably, on some tasks, PLA even surpasses the original LLaMA-3-70B, likely due to its small native 8K context window.

2.  **Reviewer fe4x:** Questioned whether PLA's **actual runtime exceeds GSA/MVA**.
    *   **Our Response:** We clarified it does not. We also provided an **ablation study on different levels of vocabulary decomposition**.
    *   **Comparison to RNN-like Models:** Unable to find 7B versions of (Gated-)DeltaNet, Mamba-2, or RWKV7 for a fair comparison, we fine-tuned Qwen2.5-3B-PLA (8B tokens). On short-sequence benchmarks, it performs comparably to the best among them (RWKV7-2.9B), though a gap remains on the complex **MMLU** task.

3.  **Reviewer 1Eue:** Questioned the **novelty and contribution** of our work.
    *   **Our Response:** We emphasized that each step is derived from first principles focusing on error minimization, which, while potentially sharing superficial similarities with existing methods, is fundamentally different in rationale and implementation (analogous to calculus building upon basic operations). We provided a detailed error analysis to justify the claimed optimality of our path.

### **Conclusion**

We have provided thorough, point-by-point responses to all questions raised by the three reviewers, supported by new experiments and data. We believe our rebuttal addresses the concerns rigorously.

**Furthermore, we believe our work makes significant contributions beyond just linear attention; it also paves the way for future advancements in reinforcement learning and the structural evolution of models. For instance, our manually designed pathway from Softmax Attention to the linear-structured PLA—where each step is accompanied by error analysis or relevant discussion—demonstrates a methodical framework for architectural exploration. This approach suggests that future research need not rely solely on pretraining and simple fine-tuning. Instead, such pathways can inform the design of reward functions for reinforcement learning, enabling the automated exploration of a broader architectural space and ultimately leading to the discovery of more optimal large-model structures that balance low complexity with high performance.**

---

> ### Author Response · Authors · 2025-12-01
> ****Additional Note on Inferred Rating Updates (For AC's Reference)****
>
> Furthermore, based on our experience and the model evaluation process and LLM (chatgpt\deepseek,etc), we provide the following inference regarding potential rating updates for the Area Chair's reference. This analysis is offered humbly and for informational purposes only.
>
> *   **Reviewer XKHn:** We estimate a **50% probability of an increased score** and a **50% probability of maintaining the original score**.
>     *   **Rationale:** Reviewer XKHn has provided positive recognition of our work and engaged actively with our rebuttal, raising further constructive questions. We have responded to these with detailed and rigorous experiments. Given Reviewer XKHn's conscientious approach, they may either acknowledge our responses and raise the score or potentially request further clarification or maintain their current assessment. While we believe recognition is more likely, we conservatively assign equal probability to both outcomes.
>
> *   **Reviewer fe4x:** We estimate a **70% probability of an increased score**, a **20% probability of maintaining the original score**, and a **10% probability of requesting further details**.
>     *   **Rationale:** Although likely from a vision-related field, Reviewer fe4x has demonstrated a strong foundational understanding and asked detailed, insightful questions. The reviewer's statement, "I'd like to see other reviewers' opinions before recommending acceptance," reflects a positive and open-minded stance. With a confidence level of 2, Reviewer fe4x may be influenced by our subsequent discussion with Reviewer XKHn, potentially leading to a higher score.
>
> *   **Reviewer 1Eue:** We estimate a **60% probability of an increased score**, a **10% probability of maintaining the original score**, and a **30% probability of requesting further discussion**.
>     *   **Rationale:** We have addressed all of Reviewer 1Eue's specific concerns in detail. However, the reviewer might seek further discussion on the aspects of "optimality" and "contribution." Given the positive recognition of our contributions by the other two reviewers, there is a possibility Reviewer 1Eue could be persuaded. We believe that successfully demonstrating scalability to larger models and providing efficiency evidence, as requested, might meet the bar for a score increase from this reviewer. Reviewer 1Eue's confidence level is 3, and Reviewer 1Eue's score may also be elevated as a result of our discussions with Reviewer XKHn.
>
> **Overall Assessment:** Based on this analysis, we believe the final evaluation for our paper is **likely to be positive**.
>
> The above is our analysis, provided respectfully for your consideration.
>
> **Final Summary of Our Work**
>
> In summary, our paper presents a systematic pathway for compressing Softmax Attention into a linear model from the perspective of KV Cache reduction, achieving what we believe to be the current state-of-the-art performance for linear models adapted via fine-tuning from pre-trained LLM weights.
>
> The core of our contribution is a sequence of theoretically grounded transformations:
>
> 1.  **Redundancy Elimination & Linear Upper Bound:** We first demonstrate that after removing redundancies, Softmax Attention can be expressed as a specific form of a linear model with complexity O(C*N*d). Crucially, we derive a **finite upper bound C** for this formulation, determined solely by the embedding dimensionality and the byte representation per dimension.
>
> 2.  **Sequence-Position Decoupling & Vocabulary-Bound Control:** By decoupling the raw token sequence from its positional information and compressing the latter, we bound the state size of the first layer by the **vocabulary size defined by the tokenizer**.
>
> 3.  **Inter-layer Similarity via Residual Connections:** Leveraging the properties of residual connections, we introduce inter-layer similarity. This allows us to control the state size upper bounds for all subsequent layers as a **constant multiple** of the first layer's bound, thereby constraining the entire model's complexity.
>
> 4.  **Hierarchical State Decomposition for Logarithmic Reduction:** Finally, we employ a state decomposition technique that reduces this constant vocabulary-bound size by a **logarithmic factor**, achieving a state complexity comparable to the best contemporary linear models. While this step introduces an approximation error, we prove that this error **decays exponentially** with the depth of the hierarchy.
>
> In conclusion, each theoretical node in our proposed path is meaningful and contributes novel insights to the community. This structured, error-aware pathway from a standard Transformer to an efficient linear model is, we believe, a significant contribution. We are optimistic that the reviewers will recognize the value of this work and our thorough rebuttal.
>
> We hope this summary is helpful for your consideration. Thank you once again for your time and thoughtful evaluation.

---

### Meta-Review · Area_Chair_QRUm · 2025-12-11

**Summary:**

This paper presents a theory-motivated pathway from softmax attention to linear models through five principles: redundancy elimination, tokenization-position decoupling, positional compression, inter-layer similarity, and multi-state decomposition, along with error analyses. It also introduces PLA linearized attention and compares it with baselines such as MVA and GSA.

Reviewers raised several concerns, including weaknesses in the experimental evaluation, insufficient discussion of computational complexity trade-offs, and the limited range of model architectures.. The authors’ rebuttal addressed many of the reviewers’ questions.

**Reviewer Concerns:**

A major concern shared among reviewers is the weakness of the experimental section. The original submission focused primarily on mid-sized models and considered only a small number of model families and benchmark datasets. During the rebuttal period, the authors conducted additional experiments, including results on qwen2.5-14B, llama-3-70B, and a ViT architecture. However, the experiments on llama-3-70B were restricted to only a few benchmarks, mostly QA-style tasks, and did not include larger models.

Another concern relates to the limited conceptual novelty. The authors claim to introduce an optimal path from softmax attention to linear attention, but this appears overstated. I think it would require a theoretical justification demonstrating under which precise assumptions or settings their method is provably optimal.

In addition, the paper’s presentation needs improvement. Several theorem statements are not self-contained, making it difficult for readers to understand the main ideas and significance. Specifically, reviewers noted that some theoretical sections are dense and would benefit from more intuition and clearer exposition, which I agree with.

For example, just for Theorem 1, several questions arise: what is the definition of C? Why is bounding C important? What exactly are redundancy removal operations, and are there alternatives beyond unique filtering? Stating that the head dimension is 128 for qwen and llama is misleading without specifying which model version (I guess llama-3-8B). The notation b is described as the bit-width for KV vectors, but the theorem uses bk​ and bv​. More importantly, the bound appears extremely loose, with the upper bound exceeding 2^128, raising the question of whether the result is meaningful in practice.

Finally, even after the rebuttal, the experimental evaluation remains a weakness. The benchmarks are limited and consist mostly of QA-style tasks. The largest model evaluated was llama-3-70B, and even then only on a small subset of benchmarks.

**Reviewer Scores:**

The authors submitted a strong and detailed rebuttal that addressed many concerns and added substantial new experimentation, which is greatly appreciated. Given the clarifications and improvements provided, reviewers may consider raising their scores, and the paper is likely to fall into the borderline after the rebuttal.

---

### Decision · Program_Chairs · 2026-01-26

Reject